

# Measurement Report: Elevated excess-NH₃ can promote the redox reaction to produce HONO: Insights from the COVID-19 pandemic

Xinyuan Zhang[1,2], Lingling Wang[3], Nan Wang[3], Shuangliang Ma[3], Shenbo Wang[2,4] *, Ruiqin Zhang[2,4]**, Dong Zhang[1,2], Mingkai Wang[2,4], Hongyu Zhang[1,2].

[1]College of Chemistry, Zhengzhou University, Zhengzhou, 450000, China

[2]Research Institute of Environmental Sciences, Zhengzhou University, Zhengzhou 450000, China

[3]Henan Provincial Ecological Environment Monitoring and Safety Center, Zhengzhou, 450000, China

[4]School of Ecology and Environment, Zhengzhou University, Zhengzhou, 450000, China

**Correspondence:** Shenbo Wang (shbwang@zzu.edu.cn) and Ruiqin Zhang (rqzhang@zzu.edu.cn)



## Abstract

The incongruity between atmospheric oxidizing capacity and $NO_x$ emissions during the COVID-19 pandemic remains puzzling. Here, we show evidence from field observations of ten sites in China that there was a noticeable increase in $NH_3$ concentrations during the COVID-19 pandemic. In addition to the meteorological conditions, the significant decrease in sulfate and nitrate concentrations enhanced the portioning of $NH_4^+$ to $NH_3$. Such conditions enable enhanced particle pH values, which in turn accelerate the redox reactions between $NO_2$ and $SO_2$ to form HONO. This mechanism partly explains the enhanced atmospheric oxidizing capacity during the pandemic and highlights the importance of coordinating the control of $SO_2$, $NO_x$, and $NH_3$ emissions.



## 1. Introduction

Atmospheric oxidizing capacity (AOC) is an important parameter that affects the formation of secondary aerosols and $O_3$ (Li et al., 2021a; Wang et al., 2023b). Identifying the factors influencing AOC is crucial for further reducing particulate matter and $O_3$ pollution. During the COVID-19 pandemic, unprecedented control measures were implemented, resulting in significant reductions in emissions from mobile traffic and stationary industry sources (Zheng et al., 2020; Wang et al., 2020a; Tian, 2020). However, studies have shown a surprising increase in AOC during this period (Huang et al., 2021a; Li et al., 2023; Liu et al., 2021; Wang et al., 2021; Zheng et al., 2020). Multiple studies have indicated that the sharp decrease in nitrogen oxide ($NO_x$) emissions leads to a substantial increase in $O_3$, as well as daytime OH and $HO_2$ radicals, and nighttime $NO_3$ radicals, subsequently resulting in an overall increase in AOC (Huang et al., 2021a; Li et al., 2023; Liu et al., 2021; Zheng et al., 2020). However, the exact relationship between $NO_x$ and AOC remains unclear.

Nitrous acid (HONO) is a critical precursor of OH radicals, contributing to more than 43–50% of OH production (Alicke, 2003) and exerting a strong influence on AOC(Zhang, 2023). Various sources of atmospheric HONO have been identified, including combustion processes (e.g., vehicle emissions) (Kramer et al., 2020; Liao et al., 2021a; Li et al., 2021b), direct emissions from soil (Su and Zhang, 2011; Oswald et al., 2013; Meusel et al., 2018), homogeneous reactions between NO and OH radicals (Pagsberg, 1997; Atkinson and Rossi, 2004), heterogeneous reactions of $NO_2$ on aerosols and ground surfaces (Zhang et al., 2020a; McFall et al., 2018; Liu et al., 2014; Liu et al., 2020a), and photolysis of nitrate (Spataro and Ianniello, 2014; Scharko et al., 2014; Romer et al., 2018; Ye et al., 2017; Shi et al., 2021). During the pandemic control periods, there was a substantial reduction in vehicle traffic flow and industrial emissions, leading to a decrease of more than 60% in $NO_x$ emissions in eastern China(Huang et al., 2021a). It was initially expected that the concentration of HONO would also decrease proportionally. However, Liu et al. observed that the decrease in HONO





concentration during the pandemic period was only 31%, which was lower than the reductions
in NO (62%) and $NO_2$ (36%) (Liu et al., 2020b). Furthermore, it is worth noting that the
concentration of HONO during the COVID-19 pandemic in 2020 was higher compared to the
levels observed during the corresponding period in 2021 (Luo et al., 2023). This finding
suggests the existence of a considerable unknown source of HONO during the COVID-19
pandemic period.
Ammonia ($NH_3$) is a significant alkaline gas in the atmosphere that plays a crucial role in
the atmospheric nitrogen cycle (Gu et al., 2022; Xu et al., 2020; Gong et al., 2011). Several
studies have indicated that $NH_3$ can promote the formation of HONO by promoting the
hydrolysis of $NO_2$ (Xu et al., 2019) or the redox reaction of $NO_2$ with $SO_2$ (Liu et al., 2023).
Moreover, previous research has shown that $NH_3$ concentrations in the atmosphere,
particularly in rural areas, significantly increased during the pandemic (Xu et al., 2022).
Consequently, the rise in $NH_3$ may contribute to the enhanced formation of HONO and
subsequently enhance AOC. Unfortunately, there is currently a lack of research on the
relationship between enhanced $NH_3$ and AOC during the COVID-19 pandemic period.
To address this, online observational data on particulate matter composition, gaseous
pollutants, and meteorological conditions from ten sites in China before and during the
COVID-19 pandemic period were analyzed to investigate the variation in $NH_3$ concentrations
and particle pH and explore the promoting effect of increased pH values on HONO formation.
To the best of our knowledge, this is the first study to discuss the reasons for the increase in
AOC during the pandemic from the perspective of the influence of $NH_3$ on HONO.

## 2. Materials and methods

### 2.1 Observation sites

Online measurements were conducted at four urban and six rural sites from January 1 to
February 29, 2020, including Sanmenxia (U-SMX), Zhoukou (U-ZK), Zhumadian (U-ZMD),



and Xinyang (U-XY), as well as rural locations including Anyang (R-AY), Xinxiang (R-XX),
Jiaozuo (R-JZ), Shangqiu (R-SQ), Nanyang (R-NY), and Puyang (R-PY). Descriptions and
the spatial distribution of these ten sites can be found in Table S1 and Figure S1 of
Supplementary Material.
**2.2 Measurements**

The aerosol and gas monitor (MARGA, Metrohm, Switzerland) was used to analyze the

hourly water-soluble ions ($Na^+$, $NH_4^+$, $K^+$, $Mg^{2+}$, $Ca^{2+}$, $Cl^-$, $NO_3^-$, and $SO_4^{2-}$) in $PM_{2.5}$, as well as
gaseous species ($NH_3$, $HNO_3$, HCl, and HONO) at ten sampling sites. The MARGA
instrument is widely used (Chen et al., 2017; Stieger et al., 2019; Twigg et al., 2022). A
detailed description of the instrument and QA/QC can be found in Text S1. In brief, the
atmospheric sample passes through a $PM_{2.5}$ cut-off head, and both particles and gases enter a
wet rotating dissolution device for diffusion. Subsequently, the particles in the sample
undergo hygroscopic growth and condensation in an aerosol supersaturated vapor generator,
followed by collection and ion chromatographic analysis. The gases in the sample are
oxidized by $H_2O_2$ in the dissolution device, absorbed into a liquid solvent, and then entered
the gas sample collection chamber for ion chromatographic quantification. The range of
minimum detection limits for water-soluble ions was between 0.002 μg/m$^3$ ($Cl^-$) to 0.081
μg/m$^3$ ($NH_4^+$). Previous works have shown that HONO observations measured using this
system agree well with other observational services, a detailed description of HONO and its
uncertainty can be found in Text S3. Overall, the limit of detection for HONO was 4 pptv and
the uncertainty was estimated to be ±20%. In addition, Uncertainties of 20% are assumed for
the detection of $NH_3$ and $NH_4^+$, while uncertainties of 10% are assumed for other
components(Wang et al., 2020b; Wang et al., 2022).
The data for $NO_2$ and $SO_2$ were obtained from a series of instruments provided by
Thermo Fisher Scientific (USA). The hourly concentrations of organic carbon (OC) in $PM_{2.5}$
were analyzed using a carbon analyzer (Model 4, Sunset Laboratory., USA). A detailed
description of the $NO_2$, $SO_2$, and carbon analyzer can be found in Text S2. The smart weather



stations (LUFFTWS500, Sutron, Germany) were utilized for synchronized observation of
meteorological parameters including temperature and relative humidity (RH).
**2.3 Data analysis.**
**2.3.1 pH prediction.**
The thermodynamic model ISORROPIA-II was used to estimate the pH value of the
particles (Fountoukis, 2007) by inputting RH, temperature, $K^+$, $Ca^{2+}$, $Mg^{2+}$, total ammonia
($TNH_x = NH_4^+ + NH_3$), total sulfuric acid ($TH_2SO_4$, $SO_4^{2-}$), total sodium (TNa, $Na^+$), total
chlorine (TCl, $Cl^-$), and total nitrate ($TNO_3 = NO_3^- + HNO_3$). The ISORROPIA model
calculated the particle hydrate ion concentration per volume of air ($H_{air}^+$) and particulate water
associated with inorganic matter ($AWC_{inorg}$). The aerosol pH value was calculated using the
following equation (Bougiatioti et al., 2016):
$$pH = -\log_{10}H_{aq}^+ = -\log_{10}\frac{1000H_{air}^+}{AWC_{inorg} + AWC_{org}} \tag{2.1}$$
where the modeled concentrations for $AWC_{inorg}$ and $H_{air}^+$ are μg/m³, and $AWC_{org}$ is the particle
water associated with the organic matters predicted using the following method:
$$AWC_{org} = \frac{m_s}{\rho_s}\frac{k_{org}}{\left(\dfrac{1}{RH} - 1\right)} \tag{2.2}$$
where $m_s$ is the mass concentrations of organic matter (OC×1.6), $\rho_s$ is the organic density
(1.35 g cm⁻³), and $k_{org}$ is the organic hygroscopicity parameter (Liu et al., 2017; Wang et al.,
2023a). $k_{org}$ is the organic hygroscopicity parameter and depends on organic functionality,
water solubility, molecular weight, and oxidation level. Han (Han et al., 2022) has reported
that the $k_{org}$ generally increased with O: C ratios, with a range of 0 – 0.3 for 23 organics,
including carboxylic acids, amino acids, sugars, and alcohols. Wang et.al(Wang et al., 2023a)
estimated that the uncertainties of $k_{org}$ value (0.06) for pH only lead to –1–3% errors, which
can be neglected. Therefore, the value of 0.06 was selected in this paper. The model has two
calculation modes: the forward mode and reverse mode, and the aerosol dissolution systems



can be set to simulate a metastable state (aqueous phase) or stable state (aqueous and solid
phase). Studies have shown that the forward mode is less affected by instrument measurement
errors than the reverse mode (Ding et al., 2019; Song et al., 2018). Additionally, the minimum
average RH of about 55% was recorded during the sampling period at the ten sites. Thus,
ISORROPIA-II was run in the forward model for the aerosol system in the metastable
condition.

**2.3.2 The HONO source analysis**

The source rates of HONO include direct emission ($P_{emi}$), homogeneous reaction of NO
and •OH ($P_{OH+NO}$), heterogeneous reaction of $NO_2$ on the ground ($P_{ground}$) and aerosol ($P_{aerosol}$),
photo-enhanced heterogeneous reaction of $NO_2$ on the ground ($P_{ground+hv}$) and aerosol
($P_{aerosol+hv}$), nitrate photolysis ($P_{nitrate}$), and unknown source ($P_{unknown}$). The detailed
methodology for its calculation is described in the Supplementary Material (**Text S4**)

**2.3.3 Production rate of HONO through redox reaction of $NO_2$ with $SO_2$.**

The redox reaction of $NO_2$ with $SO_2$ ($R_1$) is considered a crucial potential source of high
concentrations of HONO in Northern China (Cheng et al., 2019; Wang et al., 2016):

$$S(IV) + 2NO_2 + H_2O \rightarrow S(VI) + 2H^+ + 2NO_2^- \tag{R_1}$$

The rate expression for reaction ($R_1$) was estimated to:

$$d[S(VI)] / dt = k_1[NO_2][S(VI)], \tag{2.3}$$

The rate constant $k_1$ value is pH dependent, e.g., for pH, 5, $k_1 = (1.4\times10^5 + 1.24\times10^7)/2$ $M^{-1}$
$s^{-1}$. For $k_1$ values under other pH conditions and other related information, please refer to Text
S5, Table S2, and S3.



## 3. Results and discussion

### 3.1 Variations of $NH_3$, $NH_4^+$ and $TNH_x$.

The temporal variations of $NH_3$, $NH_4^+$, and $TNH_x$ at 10 sampling sites in the pre-COVID-19 pandemic period (PC, January 1 to 23, 2020) and during the COVID-19 pandemic period (DC, January 24 to February 29, 2020) are presented in Figure 1, with their average concentration listed in Table 1. In general, rural sites exhibited higher concentrations of $NH_3$, $NH_4^+$, and $TNH_x$ compared to urban sites, except for the R-NY site. This finding is consistent with previous studies conducted in Zhengzhou (Wang et al., 2020b), Shanghai (Chang et al., 2019), and Quzhou (Feng et al., 2022a), owing to the intense agricultural ammonia emissions. The highest concentrations of $NH_3$ and $TNH_x$ were recorded at site R-JZ, with average values of $25.3 \pm 11.5$ and $40.8 \pm 20.1$ $\mu g/m^3$, respectively. Site R-AY had the highest $NH_4^+$ concentration, measuring $19.3 \pm 12.9$ $\mu g/m^3$. Note that the current study area exhibited higher $NH_3$ levels compared to other regions (Table S4), which probably was attributed to the highest $NH_3$ emissions of Henan Province in China, primarily from nitrogen fertilizer application and livestock farming (Wang et al., 2018; Ma, 2020). Compared to the PC, $NH_3$ concentrations increased in the DC at all sites. Notably, rural sites experienced more significant increases in $NH_3$ concentrations than urban sites, which is similar to the trend in Shanghai (Xu et al., 2022). The largest increases in $NH_3$ concentrations were observed at R-SQ (71%, 7.3 $\mu g/m^3$) and U-ZK (37%, 4.8 $\mu g/m^3$) for rural and urban sites, respectively. In contrast, the concentrations of $NH_4^+$ and $TNH_x$ decreased in the DC with the largest reduction at rural site R-PY (51%, 12.9 $\mu g/m^3$) and urban site U-ZMD (48%, 9.3 $\mu g/m^3$). Regarding $TNH_x$, rural sites exhibited larger reductions, with site R-SQ experiencing the largest decrease of 37% (4.7 $\mu g/m^3$).

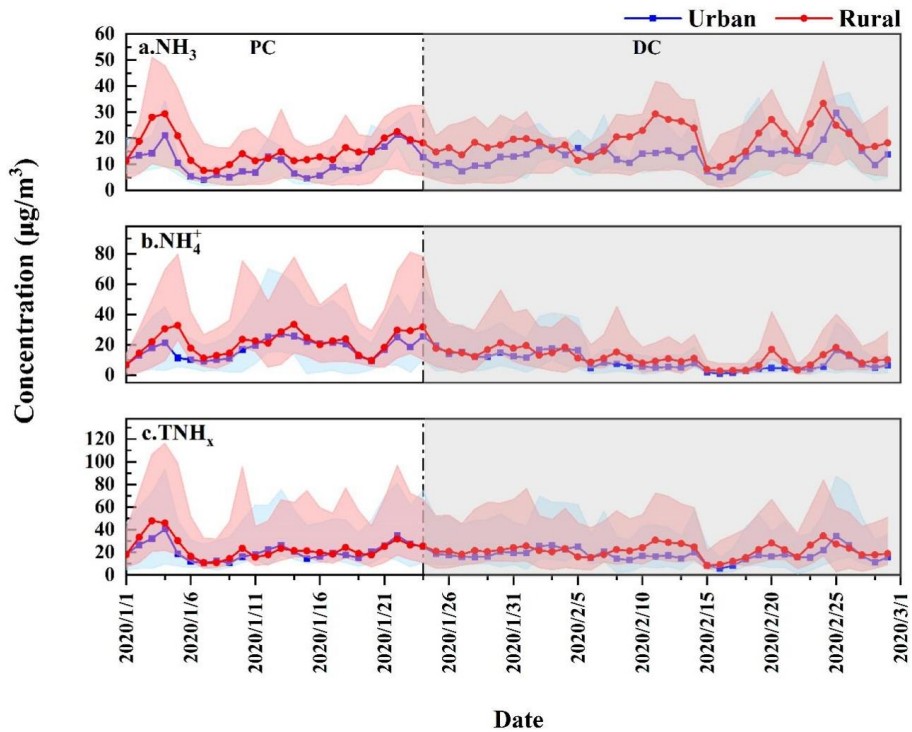


Figure 1. Temporal variations of a. $NH_3$, b. $NH_4^+$, and c. $TNH_x$ at the urban and rural sites before (PC)

and during (DC) the COVID-19 outbreak, respectively. The shaded areas of the curve represent the

maximum and minimum values.





Table 1. Changes in concentrations (mean ± standard deviation) of $NH_3$, $NH_4^+$, and $TNH_x$ at
ten sites average (overall average for the period of observation), before (PC) and during (DC)
the COVID-19 outbreak.

| Site | Substance | Average (µg/m³) | PC (µg/m³) | DC (µg/m³) |
|---|---|---|---|---|
| U-SMX | $NH_3$ | 13.8 ± 10.8 | 12.6 ± 10.1 | 14.5 ± 11.1 |
| | $NH_4^+$ | 10.9 ± 7.2 | 14.2 ± 7.2 | 8.8 ± 6.5 |
| | $TNH_x$ | 22.9 ± 14.1 | 24.9 ± 14.5 | 21.7 ± 13.8 |
| U-ZK | $NH_3$ | 15.6 ± 8.3 | 12.7 ± 6.5 | 17.4 ± 8.8 |
| | $NH_4^+$ | 13.6 ± 9.3 | 19.1 ± 8.4 | 10.3 ± 8.1 |
| | $TNH_x$ | 28.6 ± 13.7 | 30.9 ± 12.8 | 27.1 ± 14.0 |
| U-ZMD | $NH_3$ | 13.1 ± 8.4 | 11.6 ± 8.2 | 14.0 ± 8.4 |
| | $NH_4^+$ | 13.9 ± 9.8 | 19.6 ± 10.3 | 10.3 ± 7.5 |
| | $TNH_x$ | 25.7 ± 14.6 | 30.3 ± 15.1 | 22.8 ± 13.5 |
| U-XY | $NH_3$ | 7.0 ± 4.3 | 5.7 ± 4.0 | 7.9 ± 4.3 |
| | $NH_4^+$ | 11.0 ± 7.7 | 15.4 ± 7.6 | 8.3 ± 6.5 |
| | $TNH_x$ | 17.6 ± 9.8 | 20.6 ± 10.1 | 15.7 ± 9.2 |
| R-AY | $NH_3$ | 19.0 ± 8.4 | 17.9 ± 8.3 | 19.7 ± 8.4 |
| | $NH_4^+$ | 19.3 ± 12.9 | 26.4 ± 13.7 | 15.0 ± 10.3 |
| | $TNH_x$ | 36.6 ± 18.2 | 41.7 ± 20.4 | 33.4 ± 16.0 |
| R-XX | $NH_3$ | 21.7 ± 10.2 | 18.1 ± 9.3 | 23.8 ± 10.1 |
| | $NH_4^+$ | 15.9 ± 10.4 | 20.6 ± 11.0 | 13.0 ± 8.8 |
| | $TNH_x$ | 34.9 ± 17.0 | 35.1 ± 18.8 | 34.8 ± 15.8 |
| R-PY | $NH_3$ | 19.8 ± 9.4 | 16.8 ± 8.1 | 21.7 ± 9.6 |
| | $NH_4^+$ | 17.4 ± 11.8 | 25.3 ± 12.6 | 12.4 ± 8.0 |
| | $TNH_x$ | 35.2 ± 17.8 | 39.4 ± 19.8 | 32.6 ± 15.7 |
| R-JZ | $NH_3$ | 25.3 ± 11.5 | 24.1 ± 11.5 | 25.9 ± 11.4 |
| | $NH_4^+$ | 17.3 ± 11.3 | 22.7 ± 11.6 | 14.2 ± 9.9 |
| | $TNH_x$ | 40.8 ± 20.1 | 42.9 ± 22.8 | 33.5 ± 18.2 |
| R-SQ | $NH_3$ | 15.0 ± 7.9 | 10.3 ± 5.2 | 17.7 ± 7.9 |
| | $NH_4^+$ | 13.4 ± 8.5 | 18.9 ± 8.6 | 10.3 ± 6.7 |
| | $TNH_x$ | 26.3 ± 13.2 | 25.5 ± 14.0 | 26.8 ± 12.7 |
| R-NY | $NH_3$ | 5.5 ± 3.1 | 4.3 ± 2.7 | 6.2 ± 3.2 |
| | $NH_4^+$ | 10.2 ± 6.9 | 13.3 ± 7.2 | 8.4 ± 6.1 |
| | $TNH_x$ | 14.8 ± 8.5 | 16.0 ± 9.5 | 14.1 ± 7.8 |


Figure 2 illustrates the spatial distribution and the diurnal variation of $NH_3$ concentrations
in the ten sites before and during the pandemic. $NH_3$ concentrations in most sites exhibited a
unimodal trend in PC that $NH_3$ concentrations gradually increased after sunrise, reaching a
peak around noon (11:00-12:00), and then decreased to a valley around 16:00-17:00. This
diurnal pattern is similar to $NH_3$ variations observed in urban areas of Houston, USA, as a



result of the natural emissions from vegetation and soil during photosynthesis (Gong et al.,
2011). However, other studies have recorded a significant $NH_3$ peak during the early morning
of 8:00-10:00 (Ellis et al., 2011; Meng et al., 2018; Gu et al., 2022), suggesting the influence
of vehicle emissions (Gong et al., 2011; Gu et al., 2022), residual $NH_3$ mixing, soil or plant
emissions (Ellis et al., 2011), and dew volatilization (Wentworth et al., 2016; Huang et al.,
2021b). Therefore, the $NH_3$ in urban and rural areas of this study was less affected by $NH_3$
emissions from vehicles, different from the recent studies in megacities of China (e.g., Beijing
and Shanghai)(Gu et al., 2022; Wu et al., 2023; Zhang et al., 2020b). In addition to the
transport from agricultural emissions, urban $NH_3$ in this region may originate from other non-
agricultural sources, such as wastewater treatment, coal combustion, household waste, urban
green spaces, and human excrement (Chang et al., 2019).
During the COVID-19 pandemic, the diurnal variation of $NH_3$ in both urban and rural
sites still maintained a unimodal distribution. The peak values in urban sites remained
consistent with PC levels, further demonstrating that the influence of vehicles on $NH_3$ in
urban areas was limited. Notably, the peak time of $NH_3$ in rural sites shifted 1–2 hours earlier
compared to the trend in PC. Ammonia in rural areas primarily originates from nitrogen
fertilizer application, livestock, and poultry breeding (Feng et al., 2022a; Meng et al., 2018),
which are significantly influenced by temperature and RH (Liu et al., 2023). Table S5 and
Figure S2 reveal that the increased temperatures and decreased RH at rural sites in the PC,
could accelerate the evaporation of $NH_3$ and thus potentially lead to earlier peak $NH_3$
concentrations.

0

0



Figure 2. Daily variation of ammonia concentration at ten sites before (PC) and during (DC) the COVID-19 outbreak. The green dots represent the location of ten sites and their size represent the concentration of NH₃; the upper and lower whiskers represent the maximum and minimum values, respectively.



### 3.2 Gas-to-particle conversion of NH₃

Theoretically, the emissions of NH$_3$ from agricultural sources were not influenced by the containment measures, and emissions from vehicles and industries would decrease significantly in the DC. Consequently, the concentration of NH$_3$ should decrease in the DC, which was opposite to the observed trends. The decreased NH$_4^+$ in the DC suggests that the gas-particle partition of NH$_3$/ NH$_4^+$ may determine the elevated NH$_3$ concentrations. Meteorological parameters, including RH and temperature, play a crucial role in the gas-particle partitioning of NH$_3$ (Liu et al., 2023; Xu et al., 2020). Therefore, the higher temperature and lower RH in the DC (Table S5) favored the conversion of NH$_4^+$ to NH$_3$, resulting in a decrease in ε(NH$_4^+$) ([NH$_4^+$]/([NH$_3$] + [NH$_4^+$]) compared to those in the PC.

NH$_3$ primarily enters particles to neutralize acidic ions (Wang et al., 2020b; Xu et al., 2020; Liu et al., 2017; Ye et al., 2011; Wells, 1998). Accordingly, the concentrations of required ammonia (Required-NH$_x$) and excess ammonia (Excess-NH$_x$) were calculated based on the acidic substances as follows (Wang et al., 2020b):

$$\text{TNH}_x = 17 \times (\frac{[\text{NH}_4^+]}{18} + \frac{[\text{NH}_3]}{17}) \tag{3.1}$$

$$\begin{aligned}\text{Required-NH}_x = 17 \times (\frac{[\text{SO}_4^{2-}]}{48} + \frac{[\text{NO}_3^-]}{63} + \frac{[\text{Cl}^-]}{35.5} + \frac{[\text{HNO}_3]}{64} + \frac{[\text{HCl}]}{36.5}) \\ - 17 \times (\frac{[\text{Na}^+]}{23} + \frac{[\text{K}^+]}{39} + \frac{[\text{Ca}^{2+}]}{20} + \frac{[\text{Mg}^{2+}]}{12})\end{aligned} \tag{3.2}$$

$$\text{Excess-NH}_x = \text{TNH}_x - \text{Required-NH}_x \tag{3.3}$$

Where [W] represents the concentration of the substance (μg/m³).

The significant linear fitting (R$^2$ is greater than 0.96, and the slope is close to 1) in Figure S3 demonstrates that the anions and cations at each site are close to the equilibrium state. Therefore, the organic acids in PM$_{2.5}$ had less effect on NH$_3$ and NH$_4^+$, and were not considered in formula 3.2.

As shown in Figure 3 and Table S6, compared to those in the PC, the

concentration of Required-NH$_x$ in the DC significantly decreased (ranging from 37%
at site R-JZ to 58% at site R-PY), while the concentration of Excess-NH$_x$ increased
(ranging from 9% at site R-AY to 78% at site R-SQ). The reduction in the
concentrations of sulfate and nitrate (Figure S4) was responsible for the decrease in
the concentration of Required-NH$_x$. To sum up, in addition to meteorological
conditions, the substantial reduction in anthropogenic emissions of SO$_2$, NO$_x$, and
other pollutants in the DC has led to a decrease in acidic substances (e.g., sulfate and
nitrate) in particles, in turn, resulting in more gas-phase NH$_3$ concentration remaining
in the atmosphere.

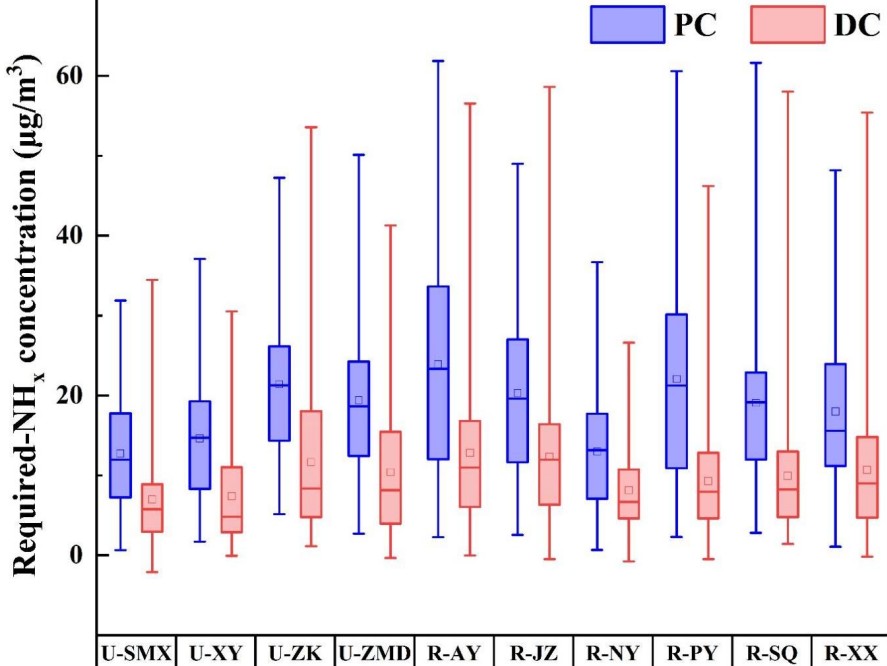


Figure 3. Box diagram of changes in Required-NH$_x$ at ten sites before (PC) and during (DC)
the COVID-19 outbreak. In each box, the top, middle, and bottom lines represent the 75, 50, and
25 percentiles of statistical data, respectively; the upper and lower whiskers represent the 90 and

10 percentiles of statistical data, respectively.



### 3.3 Particle pH before and during COVID-19

Previous studies have consistently reported that the concentration of Excess-$NH_x$ is the primary factor influencing the pH value of particles (Liu et al., 2023; Wang et al., 2020b). Therefore, the U-ZK (4.3 μg/m³) and R-PY (7.5 μg/m³) with the largest increase in Excess-$NH_x$ concentration (Table S6) for urban and rural sites respectively were selected to investigate the changes in particle pH value before and during the pandemic. The pH values at U-ZK and R-PY were 4.7 and 4.8 in the PC, respectively, which were close to the values in previous studies (Table S7). The higher pH values at the rural site than those in the urban site can be attributed to the higher concentration of Excess-$NH_x$ in rural areas.

Note that Figure 4 suggests that the particle pH increased in the DC, with an increase of 0.4 and 0.1 at U-ZK and R-PY, respectively. To explore the dominant factors that determine the local particle pH level and result in the high pH during the DC, sensitivity tests of pH to chemical species (i.e., $TNH_x$, $TH_2SO_4$, $TNO_3$, TCl, TNa, $K^+$, $Ca^{2+}$, and $Mg^{2+}$) and meteorological parameters (i.e., T and RH) were performed. A given range for a variable for U-ZK and R-PY two sites with corresponding average values of other parameters was simulated to compare its effects on pH, the input data was collected as shown in Figure 5 and Figure S5. The uncertainty of pH is shown in Figure S6. Compared to PC, even though the decrease in $TNH_x$ concentration and the increase in temperature led to a decrease in pH values (0.09 at U-ZK and 0.08 at R-PY site), this effect was outweighed by the decrease in $TH_2SO_4$ (0.07 and 0.8 at U-ZK and R-PY site, respectively) and $TNO_3$ (0.05 and 0.4 at U-ZK and R-PY site, respectively) concentrations as well as the increase in $K^+$ (0.03 at U-ZK and 0.2 at R-PY site) and $Mg^{2+}$ (0.01 at U-ZK and 0.04 at R-PY site) concentrations in the DC, and resulting in an overall increase in pH values in the DC.





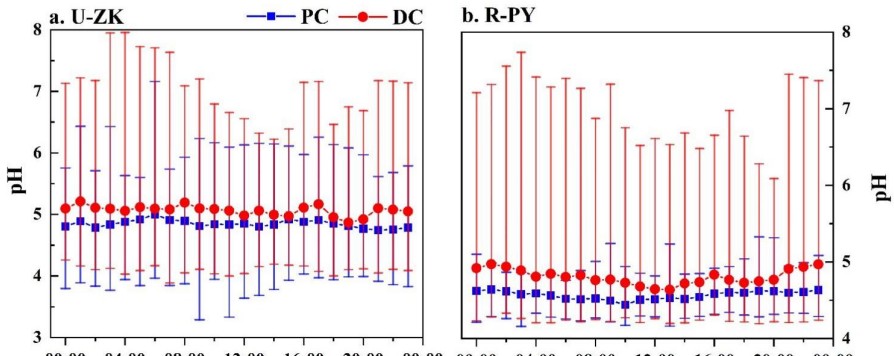

Figure 4. Daily variations of particle pH at a. U-ZK and b. R-PY sites before (PC) and during (DC) the COVID-19 outbreak. The upper and lower whiskers represent the maximum and minimum values, respectively.

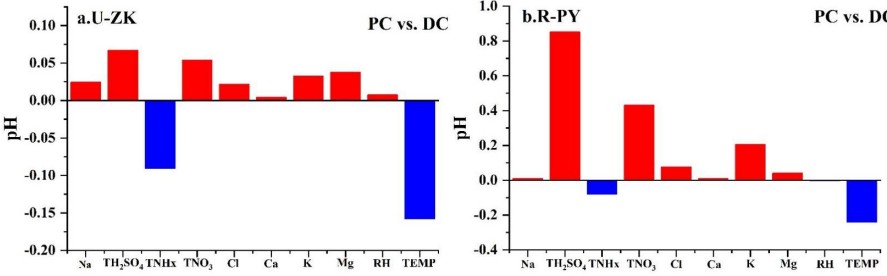

Figure 5.  Comparison of pH sensitivity (Fig. S5) to each substance by changing paraments at a. U-ZK and b. R-PY sites before (PC) and during (DC) the COVID-19 outbreak.

## 3.4 The influence of pH on HONO.

The observed HONO concentrations decreased by 18% and 54% at U-ZK (0.8 ppb) and R-PY (0.9 ppb) sites in the DC, respectively, compared to those (1.0 and 2.2 ppb) in the PC. Moreover, all the known HONO production sources rates including $P_{emi}$, $P_{OH + NO}$, $P_{ground}$, $P_{ground+hv}$, $P_{aerosol}$, $P_{aerosol+hv}$, and $P_{nitrate}$ (Figure 6) show a decreasing trend from PC to DC, with the mean reductions of 38% and 80% for U-ZK and R-PY, respectively. At the U-ZK, $P_{ground+hv}$ decreased the most (84%), while at the



R-PY, $P_{nitrate}$ had the largest decrease about 85%, which is speculated to be related to
the decrease of $NO_x$ and $NO_3^-$ concentration in DC. Note that the reduction in the
overall known source and almost individual sources were greater than the reduction in
HONO concentrations (Figure 6 and 7), thus we hypothesized that there should be
other sources capable of promoting HONO production.

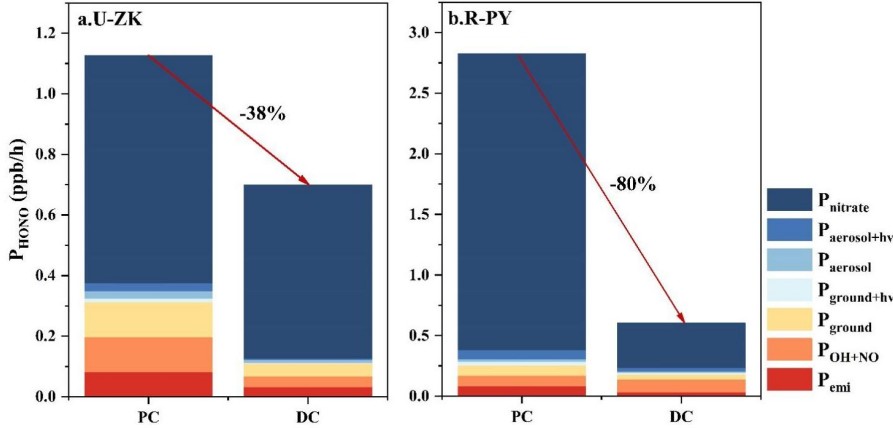


Figure 6. Comparison of HONO sources at a. U-ZK and b. R-PY sites before (PC) and during
(DC) the COVID-19 outbreak.

The relationship between HONO concentrations and other major influences at
the U-ZK and R-PY sites in the DC is displayed in Figure S7. HONO emissions were
affected by temperature to some extent(Liu et al., 2020c; Liu et al., 2020b), but there
was no significant positive correlation with temperature(Feng et al., 2022b), and
temperatures did not exceed a maximum of 10°C during this study period, suggesting
that soil emissions may not have been a major contributor to the PC HONOs during
this study period. In addition, the positive correlations between HONO with $SO_2$,
Excess-$NH_x$, $SO_4^{2-}$, and pH indicate that the $R_1$ reaction may also form an amount of
HONO and contribute to less reduction in the observed HONO concentrations.
Considering that $R_1$ mainly reacts in the liquid phase, the calculated reaction
rates of $R_1$ under the conditions of RH > 60% in the PC and DC periods are illustrated
in Figure 7. Despite the decrease in $NO_2$ and $SO_2$ concentrations in the DC, the



increase in particle pH, increasing $HSO_3^-$ concentration in the aqueous phase,
promoted the $R_1$ reaction rates by 58% and 59% at U-ZK and R-PY (Figure 7),
respectively. Consequently, the enhanced $R_1$ reaction prevented a large decrease in
HONO (18% at U-ZK and 53% at R-PY) under the conditions of a significant
reduction in vehicle emissions and a decline of 66% and 69% in $NO_2$ concentrations
at U-ZK and R-PY, respectively.

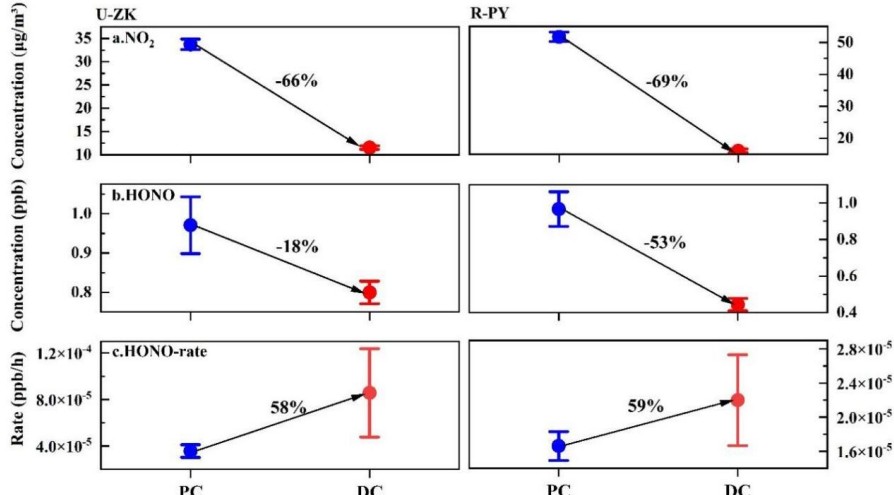


Figure 7. Decline ratios of a. $NO_2$, b. HONO concentration, and c. HONO production rate at U-
ZK and R-PY sites before (PC) and during (DC) the COVID-19 outbreak. The center point
represents the mean value, and the upper and lower whiskers represent the 95% confidence
interval of the mean.

**3.5 Uncertainty**
According to sensitivity tests of pH (Figure S5) and $R_1$ (Figure S8), pH
increases with the concentrations of cations ($TNH_x$, TNa, $K^+$, $Ca^{2+}$, and $Mg^{2+}$) and OC
increasing as well as anions ($TH_2SO_4$, $TNO_3$, and TCl) concentrations, Temp, and RH
decreasing. $R_1$ reaction rate increases with the concentrations of pH, AWC, $NO_2$, $SO_2$,
and Pressure (Pre), while increasing as well as T (K) decreasing. Therefore, two



extreme scenarios (i.e., the maximum and minimum rate scenarios) were evaluated to
estimate the uncertainty of AWC, pH, and $R_1$ HONO production rate based on the
measurement uncertainties at the U-ZK and R-PY sites. Figure S6 suggests that the
two extreme scenarios can be led to–10–7% and –71–125% uncertainties at the U-ZK
site and –10–7% and –78–123% uncertainties at the R-PY site for pH and $R_1$,
respectively.

## 339    4. Conclusions

Elevated $NH_3$ concentration was observed during the COVID-19 pandemic at
both urban and rural sites in China. In addition to the rise in temperature and decrease
in RH during the COVID-19 pandemic, which favored the conversion of $NH_4^+$ to $NH_3$,
the significant decrease in sulfate and nitrate concentrations led to the decline in
Required-$NH_x$ and was beneficial to the particle-phase $NH_4^+$ portioning to gas-phase
$NH_3$. Furthermore, under the environmental conditions of increased $NH_3$
concentration and decreased acidic substance concentration, the pH values increased
by 0.4 and 0.1 at U-ZK and R-PY increased during the pandemic, respectively.
Consequently, the high pH values accelerated the formation rate of HONO through
the oxidation-reduction reaction of $NO_2$ with $SO_2$ (an increase of 58% at U-ZK and
59% at R-PY), partially compensating for the decrease in HONO concentration and
its sources caused by the decline in vehicle emissions and $NO_2$ concentration during
the COVID-19 pandemic.

## 354    5. Implications

HONO plays a crucial role as a precursor to OH radicals in the tropospheric



atmosphere (Xue, 2022). There have been significant observations of high HONO
concentrations in urban areas during the daytime, leading to a growing interest in
understanding its sources in atmospheric chemistry (Jiang et al., 2022; Xu et al.,
2019). The heterogeneous reaction mechanism of $NO_2$ on aerosol surfaces is currently
the focus of research on HONO sources, particularly in regions with elevated levels of
atmospheric particulate matter, where it could potentially become a major contributor
to HONO production (Zhang et al., 2022; Liao et al., 2021b). One of the pathways for
heterogeneous reactions on aerosol surfaces is the redox reaction of $NO_2$ with $SO_2$.
However, the significance of this reaction in HONO production in the real atmosphere
is often overlooked, as it relies on the high pH of aerosols (Ge et al., 2019). In recent
years, there has been increasing attention on the enhancing effect of $NH_3$ on the redox
reaction, with laboratory experiments demonstrating its ability to generate substantial
amounts of HONO (Ge et al., 2019). This study highlights the importance of this
reaction based on actual atmospheric observations. Furthermore, numerous studies
have indicated that if control over $NH_3$ emissions continues to relax while $SO_2$ and
$NO_2$ emissions decrease, the particle pH in future China is expected to rise steadily
(Xie et al., 2020; Song et al., 2019; Wang et al., 2020b). Consequently, the redox
reaction of $NO_2$ with $SO_2$ could become a significant source of HONO in China,
greatly amplifying the AOC. Therefore, it is crucial to coordinate the control of $SO_2$,
$NO_x$, and $NH_3$ emissions to avoid a rapid increase in the particle pH.

**Data availability:** All the data presented in this article can be accessed through
https://zenodo.org/records/10273539. (Zhang, 2023).



**Author contributions.** XZ Data Curation, Writing - Original Draft, Visualization.
LW, NW, SM and DZ Investigation, Visualization, Data Curation. DZ, HZ and MW
Investigation. SW Conceptualization, Data Curation, Supervision. RZ Data Curation,
Funding acquisition. All people involve in discussion of the results.

**Supplement.** The supplement related to this article is available online at: XXX.

**Competing interest.** The authors declare no competing financial interest.

**Financial support.** This work was supported by the China Postdoctoral Science
Foundation (2023M733220), the Zhengzhou $PM_{2.5}$ and $O_3$ Collaborative Control and
Monitoring Project (20220347A) and the National Key Research and Development
Program of China (No. 2017YFC0212403).



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
