# Peer review of "Measurement Report: Elevated atmospheric ammonia"

_EGUsphere, 2023_

## Author Comment (AC1)

No.: egusphere-2023-2913

**Title: Measurement Report: Elevated excess-NH$_3$ can promote the redox reaction to produce HONO: Insights from the COVID-19 pandemic**

**Reviewer #2:**

**General Comments:**

This study reported that there was a noticeable increase in NH$_3$ concentrations during the COVID-19 pandemic. In addition to the meteorological conditions, the significant decrease in sulfate and nitrate concentrations enhanced the portioning of NH$_4^+$ to NH$_3$, which enables enhanced particle pH values and in turn accelerate the redox reactions between NO$_2$ and SO$_2$ to form HONO. The article has several major issues and should be considered carefully.

Thank you for your careful reading of our paper and valuable comments and suggestions. We believe that we have adequately addressed your comments. To facilitate your review, we used green highlights for your comments, yellow highlights for Reviewer #1, and red color indicating our own corrections in the manuscript.

1. In the introduction, the author comments that the exact relationship between NO$_x$, NH$_3$ and AOC remains unclear. However, it's a lengthy description of the changes in NH$_3$ and pH before and during the epidemic and there is no detailed discussion on the specific impact on AOC. In short, the research problems pointed out in the introduction have not been fully explored in the study, and many conclusions are very far-fetched.

**Response:** Thanks for your comment. In the original article, we indeed overly extended the perspectives of this study. In the revised manuscript, we removed all descriptions regarding AOC and focused on the sources of HONO, for example:

"Nitrous acid (HONO) is a critical precursor of hydroxyl radical (OH), contributing to more than 60% of OH production (Alicke, 2003; Platt et al., 1980; Kleffmann et al., 2005). The OH can react with carbon monoxide, nitrogen oxides ($NO_x$), sulfur dioxide ($SO_2$), and volatile organic compounds to produce secondary pollutants such as ozone ($O_3$) and $PM_{2.5}$ (particulate matter with an aerodynamic diameter less than or equal to 2.5 μm), thereby affecting air quality, human health, and global climate change (Li et al., 2021a; Wang et al., 2023b; Lu et al., 2018)."

2. In lines 296-297, the paper argues that HONO has other sources and that the process of $NO_2$ reacting with $SO_2$ to generate HONO is currently insufficient evidence. In addition, this reaction is affected by pH, so how much does this contribution to HONO affect atmospheric oxidation? This discussion is also sorely lacking.

**Response:** Thanks for your comment. In recent years, an increasing number of laboratory and field observation studies have shown that the reaction of $NO_2$ and $SO_2$ can generate HONO, especially under high ammonia conditions (Ge et al., 20219; Li et al., 2018; Zhang et al., 2023, 2024). Accordingly, this study found that observed $NH_3$ concentrations increased during the epidemic control period, and calculated pH values showed an increase. In addition, the positive correlations between HONO with $SO_2$, Excess-$NH_x$, $SO_4^{2-}$, and pH further indicate the existence of reaction of $NO_2$ and $SO_2$.

Moreover, we calculated the reaction rate of $NO_2$ and $SO_2$ and found that it rose by more than 50%. Although the majority of HONO unknown sources remain unexplained, this partly explains the significant decrease in $NO_x$ during the epidemic period, but the relatively low decrease in HONO concentrations.

Ge, S., Wang, G., Zhang, S., Li, D., and Zhang, H.: Abundant $NH_3$ in China enhances atmospheric HONO production by promoting the heterogeneous reaction of $SO_2$ with $NO_2$. Environ. Sci. Technol. 53, 14339 – 14347, https://doi.org/10.1021/acs.est.9b04196, 2019.

Li, L., Hoffmann, M. R., and Colussi, A. J.: Role of nitrogen dioxide in the production of sulfate during Chinese haze-aerosol episodes, Environ. Sci. Technol., 52, 2686 – 2693, https://doi.org/10.1021/acs.est.7b05222, 2018.

Zhang, X., Tong, S., Jia, C., Zhang, W., Wang, Z., Tang, G., Hu, B., Liu, Z., Wang, L., Zhao, P., Pan, Y., and Ge, M.: Elucidating HONO formation mechanism and its essential contribution to OH during haze events., npj. ClWim. Atmos. Sci., 6, 55, https://doi.org/10.1038/s41612-023-00371-w, 2023.

Zhang, P., Li, H., Ma, Q., Chen, T., Chu, B., Yu, Y., and He, H.: $SO_2$ photoaging enhances the surface conversion of $NO_2$-to-HONO on elemental carbon, Environ. Sci. Technol. Lett., 11, 143 – 149, https://doi.org/10.1021/acs.estlett.3c00878, 2024.

3. About HONO sources calculation, there are also many issues. The emission of motor vehicles at different stations varies greatly, so it is unreasonable to use 0.65% as the emission factor of HONO at all stations.

**Response:** Thanks for your comments.

Firstly, We determined whether it is necessary to calculate vehicle emissions, and a supplementary HONO emission factor table from vehicle emissions was added to support the selection of factors in the revised version:

"HONO can be released directly into the atmosphere through vehicle exhaust (Burling et al., 2010; Veres et al., 2010). The lifetime of HONO in the atmosphere is relatively short, so vehicle emissions significantly contribute to urban atmospheric HONO (Chen et al., 2023; Liu et al., 2021a). Considering that there has been a significant reduction in vehicle emissions in urban areas during DC. Additionally, the R-PY site is far from roads. Thus, vehicle emissions may not be the primary source of HONO for the U-ZK site during DC and R-PY sites during entire periods. To further validate the above conclusions, the conditional bivariate probability function diagrams of $NO_2$ at U-ZK and R-PY sites during PC and DC are depicted in Figure S2. $NO_2$ predominantly originated from long-distance transport at the U-ZK site during DC and the R-PY site during both PC and DC. Consequently, vehicle emissions are only calculated for the U-ZK site during the PC.

Here we use the $HONO/NO_x$ ratio to estimate HONO concentration, which is generally considered to be the vehicle emission factor (Kramer et al., 2020; Hao et al., 2020; Yu et al., 2022) for HONO. The calculation formula is as follows:

$$[HONO_{emi}] = 0.8\% \times [NO_x] \tag{1}$$

where $[HONO_{emi}]$ and $[NO_x]$ represent the HONO concentration emitted by vehicles and the observed $NO_x$ concentration, respectively. Regarding previous studies (Table S3), 0.8% was selected as the vehicle emission factor, considering differences in vehicle type, fuel composition, and other factors (Kramer et al., 2020; Hao et al., 2020; Huang et al., 2017)."

[Figure]

Figure S2. Result of conditional bivariate probability function plots: $NO_2$ at U-ZK and R-PY sites before (PC) and during (DC) the COVID-19 outbreak. The color scale bar represents $NO_2$ concentration.

Table S3. Summary of vehicle emission factors.

| Observation site | Period | Emission factor (%) | Reference |
|---|---|---|---|
| Beijing | 2020 | 0.79 | (Meng et al., 2020) |
| Hong Kong | 2015 | 0.4–1.8 | (Yun et al., 2017) |
| Hong Kong | 2011 | 0.5–1.6 | (Xu et al., 2015) |
| Kiesberg Tunnel | 2001 | 0.8 | (Kleffmann et al., 2003) |
| Kiesberg Tunnel | 1997 | 0.3–0.8 | (Kurtenbach and Wiesen, 2001) |
| Guangzhou | 2019 | 1.31 | (Li et al., 2021b) |

Secondly, the sources of HONO were recalculated to better investigate the changes in HONO between PC and DC periods:

**Text S4 Sources of HONO**

4.1 Direct emission

HONO can be released directly into the atmosphere through vehicle exhaust (Burling et al., 2010; Veres et al., 2010). The lifetime of HONO in the atmosphere is relatively short, so vehicle emissions significantly contribute to urban atmospheric HONO (Chen et al., 2023; Liu et al., 2021a). Considering that there has been a significant reduction in vehicle emissions in urban areas during DC. Additionally, the R-PY site is far from roads. Thus, vehicle emissions may not be the primary source of HONO for the U-ZK site during DC and R-PY sites during entire periods. To further validate the above conclusions, the conditional bivariate probability function diagrams of $NO_2$ at U-ZK and R-PY sites during PC and DC are depicted in Figure S2. $NO_2$ predominantly originated from long-distance transport at the U-ZK site during DC and the R-PY site during both PC and DC. Consequently, vehicle emissions are only calculated for the U-ZK site during the PC.

Here we use the $HONO/NO_x$ ratio to estimate HONO concentration, which is generally considered to be the vehicle emission factor (Kramer et al., 2020; Hao et al., 2020; Yu et al., 2022) for HONO. The calculation formula is as follows:

$$[HONO_{emi}] = 0.8\% \times [NO_x] \tag{1}$$

where [$HONO_{emi}$] and [$NO_x$] represent the HONO concentration emitted by vehicles and the observed $NO_x$ concentration, respectively. Regarding previous studies (Table S3), 0.8% was selected as the vehicle emission factor, considering differences in vehicle type, fuel composition, and other factors (Kramer et al., 2020; Hao et al., 2020; Huang et al., 2017).

**4.2 Homogeneous reaction of NO and •OH**

The reaction between NO and •OH is the primary gas-phase reaction source of HONO at high NO concentrations, and the production rate contribution ($P_{OH+NO}$) for this reaction can be calculated as:

$$P_{OH+NO} = k_{OH+NO}[OH][NO] \tag{2}$$

where $k_{OH+NO}$ ($7.2 \times 10^{-12}$ cm$^3$ molecule$^{-1}$ s$^{-1}$) is the rate constant for the reactions at 298K (Li et al., 2012). •OH concentration was simulated according to the empirical model (Hu et al., 2022; Wang et al., 2025):

$$[OH] = 4.1 \times 10^9 \times \frac{J(O^1D) \times J(NO_2) \times (140 \times [NO_2] + 1) + [HONO] \times J(HONO)}{0.41 \times [NO_2]^2 + 1.7 \times [NO_2] + 1 + [NO] \times k_{NO+OH} + [HONO] \times k_{NO+OH}} \tag{3}$$

where, J ($O^1$ D), J ($NO_2$), and J (HONO) are the photolysis rates calculated using the TUV model (v5.2; available at http://cprm.acom.ucar.edu/Models/TUV/). The cloud optical depth value for the TUV model was adjusted so that the predicted UVB radiation intensity matched the observations (Lyu et al., 2019; Wang et al., 2022b). The calculated •OH concentration varied from $0.1 \times 10^6$ to $4 \times 10^6$ molecule/cm$^3$ at U-ZK and $0.1 \times 10^6$ to $5 \times 10^6$ molecule/cm$^3$ t R-PY, which was comparable to the levels in other cities of North China (Li et al., 2018; Fuchs et al., 2017; Yang et al., 2017). Since there is no photolysis at night, the •OH concentration was assumed to be $0.8 \times 10^6$ molecule/cm$^3$ (Wang et al., 2022).

**4.3 Heterogeneous conversion of $NO_2$ to HONO**

**4.3.1 Heterogeneous dark reactions**

The heterogeneous conversion of $NO_2$ to HONO on the ground ($P_{ground}$) and on the aerosol surface ($P_{aerosol}$) was calculated based on parameters obtained from experiments or observations.

$$P_{ground} = \frac{1}{8}\gamma_1 \times [NO_2] \times C_{NO_2} \times \frac{S_g}{V} \tag{4}$$

$$P_{aerosol} = \frac{1}{4}\gamma_2 \times [NO_2] \times C_{NO_2} \times \frac{S_a}{V} \tag{5}$$

$$\frac{S_g}{V} = \frac{1}{MLH} \tag{6}$$

$$C_{NO_2} = \sqrt{\frac{8RT}{\pi M}} \tag{7}$$

where $C_{NO2}$ is the average molecular velocity of $NO_2$ molecule (m s$^{-1}$); R is the ideal gas constant; T is the temperature (K); M is the molecular weight of $NO_2$ (kg mol$^{-1}$); MLH is the height of the mixed layer, which is determined to be 50 m due to HONO formation on the ground level and its short lifetime (Liu et al., 2020b); $S_a/V$ is the surface area to volume ratio of aerosol, estimated by Su et al. (Su et al., 2008).

**4.3.2 Heterogeneous photo-enhanced reactions**

The heterogeneous photo-enhanced reactions of $NO_2$ on the surface of the ground ($P_{ground + hv}$) and the surface of the aerosol ($P_{aerosol + hv}$) were calculated following (Zhang et al., 2020a):

$$P_{ground+hv} = \frac{1}{8} \times C_{NO_2} \times \frac{1}{MLH} \times \gamma_1 \times \frac{J_{NO_2}}{J_{NO_2,noon}} \times [NO_2] \qquad (8)$$

$$P_{aerosol+hv} = \frac{1}{4} \times C_{NO_2} \times \frac{S_a}{V} \times \gamma_2 \times \frac{J_{NO_2}}{J_{NO_2,noon}} \times [NO_2] \qquad (9)$$

where $J_{NO_2}$ and $J_{NO_2, noon}$ are the photolysis rate of $NO_2$ and the photolysis rate of $NO_2$ at noon during the day, respectively.

$\gamma_1$ and $\gamma_2$ are the absorption coefficient of $NO_2$ on the ground and aerosol surface, respectively, which is assumed to be $4 \times 10^{-6}$ (Yu et al., 2022; Zhang et al., 2021; Zhang et al., 2020a). Moreover, we discuss the uncertainties based on the recommended values of $2 \times 10^{-6}$–$1 \times 10^{-5}$ as upper and lower bounds(Chen et al., 2023; VandenBoer et al., 2013; Wong et al., 2011). Results show (Figure S3) that the uncertainties for $P_{ground}$, $P_{aerosol}$, $P_{groung+hv}$, and $P_{aerosol+hv}$ are −50% to 150%, −50% to 151%, −20% to 120%, and −50% to 121% at the U-ZK, respectively. At the R-PY, the uncertainties for $P_{ground}$, $P_{aerosol}$, $P_{groung+hv}$, and $P_{aerosol+hv}$ are −50% to 150%, −50% to 151%, −20% to 120%, and −50% to 121%, respectively.

4.4 Nitrate photolysis

The nitrate photolysis ($P_{nitrate}$) was calculated based on the measured nitrate concentration ($NO_3^-$) from $PM_{2.5}$ and nitrate photolysis rate ($J_{nitrate \rightarrow HONO}$):

$$P_{nitrate} = J_{nitrate \rightarrow HONO} \times [NO_3^-] \qquad (10)$$

where the $J_{nitrate \rightarrow HONO}$ was simulated by normalizing UV values, when the Zenit Angle is 0°, $J_{nitrate \rightarrow HONO}$ varied within the range of $1.22 \times 10^{-5}$ to $4.84 \times 10^{-4}$ s$^{-1}$, with an average value of $8.24 \times 10^{-5}$ s$^{-1}$ (Bao et al., 2018)."

Unfortunately, for MLH, $S_a/V$, and the relationship between k1 and temperature, as there were no observational data or scientifically established estimation methods, this study did not consider their variations. This omission may lead to differences in conclusions and warrants further investigation in future research.

[Figure]

Figure S2. Result of conditional bivariate probability function plots: $NO_2$ at U-ZK and R-PY sites before (PC) and during (DC) the COVID-19 outbreak. The color scale bar represents $NO_2$ concentration.

[Figure]

Figure S3. HONO production rate using different uptake rates of $NO_2$ at the U-ZK and R-PY sites before (PC) and during (DC) the COVID-19 outbreak. (a)$P_{ground}$, (b) $P_{aerosol}$, (c) $P_{ground+hv}$, and (d) $P_{aerosol+hv}$

Table S3. Summary of vehicle emission factors.

| Observation site | Period | Emission factor (%) | Reference |
|---|---|---|---|
| Beijing | 2020 | 0.79 | (Meng et al., 2020) |
| Hong Kong | 2015 | 0.4–1.8 | (Yun et al., 2017) |
| Hong Kong | 2011 | 0.5–1.6 | (Xu et al., 2015) |
| Kiesberg Tunnel | 2001 | 0.8 | (Kleffmann et al., 2003) |
| Kiesberg Tunnel | 1997 | 0.3–0.8 | (Kurtenbach and Wiesen, 2001) |
| Guangzhou | 2019 | 1.31 | (Li et al., 2021b) |

4. The uptake coefficient of $NO_2$ on surfaces is not mentioned.

**Response:** Thank you for your comment. We have added the description of the uptake coefficient of $NO_2$:

"$\gamma_1$ and $\gamma_2$ are the absorption coefficient of $NO_2$ on the ground and aerosol surface, respectively, which is assumed to be $4 \times 10^{-6}$ (Yu et al., 2022; Zhang et al., 2021; Zhang et al., 2020a). Moreover, we discuss the uncertainties based on the recommended values of $2 \times 10^{-6}$–$1 \times 10^{-5}$ as upper and lower bounds(Chen et al., 2023; VandenBoer et al., 2013; Wong et al., 2011). Results show (Figure S3) that the uncertainties for $P_{ground}$, $P_{aerosol}$, $P_{groung+hv}$, and $P_{aerosol+hv}$ are −50% to 150%, −50% to 151%, −20% to 120%, and −50% to 121% at the U-ZK, respectively. At the R-PY, the uncertainties for $P_{ground}$, $P_{aerosol}$, $P_{groung+hv}$, and $P_{aerosol+hv}$ are −50% to 150%, −50% to 151%, −20% to 120%, and −50% to 121%, respectively."

[Figure]

Figure S3. HONO production rate using different uptake rates of NO$_2$ at the U-ZK and R-PY sites before (PC) and during (DC) the COVID-19 outbreak. (a)P$_{ground}$, (b) P$_{aerosol}$, (c) P$_{ground+hv}$, and (d) P$_{aerosol+hv}$

5. The same OH concentration used at all station is also controversial.

**Response:** Thank you for your comments. We have modified the method for determining •OH concentration in the revised manuscript:

•OH concentration was simulated according to the empirical model (Hu et al., 2022; Wang et al., 2025):

$$[OH] = 4.1 \times 10^9 \times \frac{J(O^1D) \times J(NO_2) \times (140 \times [NO_2] + 1) + [HONO] \times J(HONO)}{0.41 \times [NO_2]^2 + 1.7 \times [NO_2] + 1 + [NO] \times k_{NO+OH} + [HONO] \times k_{NO+OH}} \quad (3)$$

where, $J(O^1D)$, $J(NO_2)$, and $J(HONO)$ are the photolysis rates calculated using the TUV model (v5.2; available at http://cprm.acom.ucar.edu/Models/TUV/). The cloud optical depth value for the TUV model was adjusted so that the predicted UVB radiation intensity matched the observations (Lyu et al., 2019; Wang et al., 2022b). The calculated •OH concentration varied from $0.1 \times 10^6$ to $4 \times 10^6$ molecule/cm$^3$ at U-ZK and $0.1 \times 10^6$ to $5 \times 10^6$ molecule/cm$^3$ t R-PY, which was comparable to the levels in other cities of North China (Li et al., 2018; Fuchs et al., 2017; Yang et al., 2017). Since there is no photolysis at night, the •OH concentration was assumed to be $0.8 \times 10^6$ molecule/cm$^3$ (Wang et al., 2022).

6. In the supplement, is the equation (4) utilized in the calculation?

**Response:** Sorry for the mistake. We have corrected the formula:

$$P_{ground} = \frac{1}{8} \gamma_1 \times [NO_2] \times C_{NO_2} \times \frac{S_g}{V} \quad (4)$$

$$P_{aerosol} = \frac{1}{4} \gamma_2 \times [NO_2] \times C_{NO_2} \times \frac{S_a}{V} \quad (5)$$

$$\frac{S_g}{V} = \frac{1}{MLH} \tag{6}$$

7. The $J_{HONO}$ and $J_{nitrate}$ used are suggested to be described in detail.

**Response:** Thanks for your suggestion. We have added a detailed description:

"$J(O^1D)$, $J(NO_2)$, and $J(HONO)$ are the photolysis rates calculated using the TUV model (v5.2; available at http://cprm.acom.ucar.edu/Models/TUV/). The cloud optical depth value for the TUV model was adjusted so that the predicted UVB radiation intensity matched the observations (Lyu et al., 2019; Wang et al., 2022b)."

"The $J_{nitrate \rightarrow HONO}$ was simulated by normalizing UV values when the Zenit Angle is 0°, $J_{nitrate \rightarrow HONO}$ varied within 
[revised manuscript text omitted]
. The detection limit is about 0.03 $\mu g/m^3$ (Mikuska et al., 2008; Zhao et al., 2010). In addition, HONO observations measured with this AIM-IC system agree well with HONO observations measured with the other systems (VandenBoer et al., 2014). Therefore, it is feasible to measure HONO using this instrument.

**Text S3 Detailed description of the NO$_2$、SO$_2$ and carbon analyzer.**

The NO$_2$ analyzer utilized the chemiluminescence technique to measure the concentration of NO$_2$ in the air. This involved converting NO$_2$ to NO using a molybdenum converter, and then quantifying the NO concentration. The principle behind the SO$_2$ analyzer involved measuring the amount of ultraviolet light emitted during the decay of high-energy state SO$_2$. This emitted light was used to calculate the concentration of SO$_2$.

The carbon analyzer principle is primarily based on the NIOSH-5040 method, which involves analyzing the thermal optical transmittance of quartz filter samples. It employs a calibrated non-dispersive infrared sensor to detect the evolving carbon. Under controlled conditions with inert helium gas, carbon formed during a gradually increasing temperature gradient is referred to as OC, while carbon evolved under a mixture of 90% helium.

**Text S4 Sources of HONO**

4.1 Direct emission

HONO can be released directly into the atmosphere through vehicle exhaust (Burling et al., 2010; Veres et al., 2010). The lifetime of HONO in the atmosphere is relatively short, so vehicle emissions significantly contribute to urban atmospheric HONO (Chen et al., 2023; Liu et al., 2021a). Considering that there has been a significant reduction in vehicle emissions in urban areas during DC. Additionally, the R-PY site is far from roads. Thus, vehicle emissions may not be the primary source of HONO for the U-ZK site during DC and R-PY sites during entire periods. To further validate the above conclusions, the conditional bivariate probability function diagrams of $NO_2$ at U-ZK and R-PY sites during PC and DC are depicted in Figure S2. $NO_2$ predominantly originated from long-distance transport at the U-ZK site during DC and the R-PY site during both PC and DC. Consequently, vehicle emissions are only calculated for the U-ZK site during the PC.

Here we use the HONO/$NO_x$ ratio to estimate HONO concentration, which is generally considered to be the vehicle emission factor (Kramer et al., 2020; Hao et al., 2020; Yu et al., 2022) for HONO. The calculation formula is as follows:

$$[HONO_{emi}] = 0.8\% \times [NO_x] \tag{1}$$

where [$HONO_{emi}$] and [$NO_x$] represent the HONO concentration emitted by vehicles and the observed $NO_x$ concentration, respectively. Regarding previous studies (Table S3), 0.8% was selected as the vehicle emission factor, considering differences in vehicle type, fuel composition, and other factors (Kramer et al., 2020; Hao et al., 2020; Huang et al., 2017).

4.2 Homogeneous reaction of NO and •OH

The reaction between NO and •OH is the primary gas-phase reaction source of HONO at high NO concentrations, and the production rate contribution ($P_{OH+NO}$) for this reaction can be calculated as:

$$P_{OH+NO} = k_{OH+NO}[OH][NO] \tag{2}$$

where $k_{OH+NO}$ ($7.2 \times 10^{-12}$ cm$^3$ molecule$^{-1}$ s$^{-1}$) is the rate constant for the reactions at

298K (Li et al., 2012). •OH concentration was simulated according to the empirical model (Hu et al., 2022; Wang et al., 2025):

$$[OH] = 4.1 \times 10^9 \times \frac{J(O^1D) \times J(NO_2) \times (140 \times [NO_2] + 1) + [HONO] \times J(HONO)}{0.41 \times [NO_2]^2 + 1.7 \times [NO_2] + 1 + [NO] \times k_{NO+OH} + [HONO] \times k_{NO+OH}} \tag{3}$$

where, J (O$^1$ D), J (NO$_2$), and J (HONO) are the photolysis rates calculated using the

TUV model (v5.2; available at http://cprm.acom.ucar.edu/Models/TUV/). The cloud optical depth value for the TUV model was adjusted so that the predicted UVB radiation intensity matched the observations (Lyu et al., 2019; Wang et al., 2022b). The calculated

•OH concentration varied from $0.1 \times 10^6$ to $4 \times 10^6$ molecule/cm$^3$ at U-ZK and $0.1 \times$

$10^6$ to $5 \times 10^6$ molecule/cm$^3$ t R-PY, which was comparable to the levels in other cities of North China (Li et al., 2018; Fuchs et al., 2017; Yang et al., 2017). Since there is no photolysis at night, the •OH concentration was assumed to be $0.8 \times 10^6$ molecule/cm$^3$

(Wang et al., 2022).

4.3 Heterogeneous conversion of NO$_2$ to HONO

4.3.1 Heterogeneous dark reactions

The heterogeneous conversion of NO$_2$ to HONO on the ground (P$_{ground}$) and on the aerosol surface (P$_{aerosol}$) was calculated based on parameters obtained from experiments or observations.

$$P_{ground} = \frac{1}{8}\gamma_1 \times [NO_2] \times C_{NO_2} \times \frac{S_g}{V} \tag{4}$$

$$P_{aerosol} = \frac{1}{4}\gamma_2 \times [NO_2] \times C_{NO_2} \times \frac{S_a}{V} \tag{5}$$

$$\frac{S_g}{V} = \frac{1}{MLH} \tag{6}$$

$$C_{NO_2} = \sqrt{\frac{8RT}{\pi M}} \tag{7}$$

where $C_{NO2}$ is the average molecular velocity of $NO_2$ molecule (m s$^{-1}$); R is the ideal gas constant; T is the temperature (K); M is the molecular weight of $NO_2$ (kg mol$^{-1}$);

MLH is the height of the mixed layer, which is determined to be 50 m due to HONO

formation on the ground level and its short lifetime (Liu et al., 2020b); $S_a$/V is the surface area to volume ratio of aerosol, estimated by Su et al. (Su et al., 2008).

4.3.2 Heterogeneous photo-enhanced reactions

The heterogeneous photo-enhanced reactions of $NO_2$ on the surface of the ground ($P_{ground + hv}$) and the surface of the aerosol ($P_{aerosol + hv}$) were calculated following (Zhang et al., 2020a):

$$P_{ground+hv} = \frac{1}{8} \times C_{NO_2} \times \frac{1}{MLH} \times \gamma_1 \times \frac{J_{NO_2}}{J_{NO_2,noon}} \times [NO_2] \tag{8}$$

$$P_{aerosol+hv} = \frac{1}{4} \times C_{NO_2} \times \frac{S_a}{V} \times \gamma_2 \times \frac{J_{NO_2}}{J_{NO_2,noon}} \times [NO_2] \tag{9}$$

where $JNO_2$ and $JNO_{2, noon}$ are the photolysis rate of $NO_2$ and the photolysis rate of $NO_2$

at noon during the day, respectively.

$\gamma_1$ and $\gamma_2$ are the absorption coefficient of $NO_2$ on the ground and aerosol surface, respectively, which is assumed to be $4 \times 10^{-6}$ (Yu et al., 2022; Zhang et al., 2021; Zhang et al., 2020a). Moreover, we discuss the uncertainties based on the recommended values of $2 \times 10^{-6}$–$1 \times 10^{-5}$ as upper and lower bounds(Chen et al., 2023; VandenBoer et al.,

2013; Wong et al., 2011). Results show (Figure S3) that the uncertainties for $P_{ground}$,

$P_{aerosol}$, $P_{groung+hv}$, and $P_{aerosol+hv}$ are −50% to 150%, −50% to 151%, −20% to 120%, and

−50% to 121% at the U-ZK, respectively. At the R-PY, the uncertainties for $P_{ground}$,

$P_{aerosol}$, $P_{groung+hv}$, and $P_{aerosol+hv}$ are −50% to 150%, −50% to 151%, −20% to 120%, and

−50% to 121%, respectively.

4.4 Nitrate photolysis

The nitrate photolysis ($P_{nitrate}$) was calculated based on the measured nitrate concentration ($NO_3^-$) from $PM_{2.5}$ and nitrate photolysis rate ($J_{nitrate \rightarrow HONO}$):

$$P_{nitrate} = J_{nitrate \rightarrow HONO} \times [NO_3^-] \tag{10}$$

where the $J_{nitrate \rightarrow HONO}$ was simulated by normalizing UV values, when the Zenit Angle is 0°, $J_{nitrate \rightarrow HONO}$ varied within the range of $1.22 \times 10^{-5}$ to $4.84 \times 10^{-4}$ s$^{-1}$, with an average value of $8.24 \times 10^{-5}$ s$^{-1}$ (Bao et al., 2018).

**Text S5 Estimation of HONO formation rate**

The redox reaction of $NO_2$ with $SO_2$ ($R_1$) is considered a crucial potential source of high concentrations of HONO in Northern China (Wang et al., 2016b; Cheng, 2016):

$$S(IV) + 2NO_2 + H_2O \rightarrow S(IV) + 2H^+ + 2NO_2^- \tag{R_1}$$

The rate expression for the reaction was estimated to:

$$d[S(VI)]/dt = k_1[NO_2][S(VI)],\qquad(11)$$

where the $k_1 = (1.4\times10^5 + 1.24\times10^7)/2\ M^{-1}s^{-1}$ for the pH range < 5;

$k_1 = (23.25\times(pH–5) + 1.4 + 124)/2\times10^5\ M^{-1}s^{-1}$ for the pH range 5 < pH < 5.3;

$k_1 = (23.25\times(pH–5) + 1.4 + 12.6\times(pH–5.3) + 124)/2\times10^5\ M^{-1}s^{-1}$ for the pH range 5.3 <

pH < 5.8;

$k_1 = (12.6\times(pH–5.3) + 124+20)/2\times10^5\ M^{-1}s^{-1}$ for the pH range 5.8 < pH < 8.7; and $k_1 = (2\times10^6 + 1.67\times10^7)/2\ M^{-1}s^{-1}$ for the pH range pH > 8.7. (Seinfeld et al., 1998)

In the above calculation formulas, the concentration of gas in the liquid is determined by Henry's constant ($H^*$). The calculation formula is in Table S4. $SO_2$ has a dissociation equilibrium in the solution, producing $HSO_3^-$ and $SO_3^{2-}$. The ionization constants (K)

are shown in the following Table S5. The H* and K are temperature-dependent. The values are given in Tables S4 and S5 under the condition of 298K, converted to the value under the actual temperature using the following calculation formula:

$$H(T)\ or\ K(T) = H(T_{298K})\ or\ K(T_{298K})\exp\left[-\frac{\Delta H_{298K}}{R}\left(\frac{1}{T}-\frac{1}{298K}\right)\right]\qquad(12)$$

where H(T)、K(T)、$H(T_{298K})$, and $K(T_{298K})$ represent the H* and K at actual temperature and 298 K, respectively.

Influences of ionic strength on $R_1$ were not considered because of the high values predicted by the ISORROPIA-II model during the sampling periods (Cheng et al.,

2016). To evaluate the effects of mass transport, the formulation of a standard resistance model was adopted:

$$\frac{1}{R_{H,aq}} = \frac{1}{R_{aq}} + \frac{1}{J_{aq,lim}}$$
(13)

where $R_{H,aq}$ is the sulfate production rate, $R_{aq}$ is the aqueous-phase reaction rate and

$J_{aq,lim}$ is the limiting mass transfer rate. which could be calculated by the formulas as follows:

$$J_{aq,lim} = \min\{J_{aq}(SO_2), J_{aq}(X)\}$$
(14)

$$J_{aq}(X) = k_{MT}(X)\cdot[X]$$
(15)

where [X] refers to the aqueous-phase concentrations of $SO_2$ or the oxidants $O_{xi}$

calculated by the equation in Table S4. The mass transfer rate coefficient $k_{MT}(X)$ ($s^{-1}$)

can be calculated by:

$$k_{MT} = [\frac{R_p^2}{3D_g} + \frac{4R_p}{3\alpha v}]^{-1}$$
(16)

where $R_p$ is the aerosol radius, $D_g$ is the gas-phase molecular diffusion coefficient (0.2

$cm^2$ $s^{-1}$ at 293K), $v$ is the mean molecular speed of X ($3\times10^4$ cm $s^{-1}$), and $a$ is the mass accommodation of X on the droplet surface, and we adopted values of 0.11 and $2E^{-4}$ for

$SO_2$ and $NO_2$, respectively referring to Cheng et al. (Cheng, 2016).

**Figures**

[Figure]

Figure S1. Sampling point map in Henan Province, China. © 2019 National Geomatics

Center of China. i.e., urban sites at Sanmenxia (U-SMX), Zhoukou (U-ZK), Zhuamdian (U-ZMD) and Xinyang (U-XY), rural sites at Anyang (R-AY), Xinxiang (R-XX),

Puyang (R-PY), Jiaozuo(R-JZ), Shangqiu (R-SQ) and Nanyang (R-NY). All rights reserved.

[Figure]

Figure S2. Result of conditional bivariate probability function plots: $NO_2$ at U-ZK and R-PY sites before (PC) and during (DC) the COVID-19 outbreak. The color scale bar represents $NO_2$ concentration.

[Figure]

Figure S3. HONO production rate using different uptake rates of $NO_2$ at the U-ZK and R-PY sites before (PC) and during (DC) the COVID-19 outbreak. (a)$P_{ground}$, (b) $P_{aerosol}$, (c) $P_{ground+hv}$, and (d) $P_{aerosol+hv}$

[Figure]

Figure S4. Daily changes in temperature and relative humidity (RH) in rural sites before (PC) and during (DC) the COVID-19 outbreak, the error bar represents the standard deviation. The upper and lower whiskers represent the standard deviation.

[Figure]

Figure S5. The equilibrium state of anions and cations at ten sites before (PC) and during (DC) the COVID-19 outbreak.

[Figure]

Figure S6. Concentrations of the water-soluble ions at the ten sites before (PC) and during (DC) the COVID-19 outbreak.

[Figure]

Figure S7. Sensitivity tests of pH to each factor. The vertical bar represents the mean values before (PC) and during (DC) the COVID-19 outbreak. A given range for a variable (i.e., $TNH_x$) with corresponding average values of other parameters (i.e., $TH_2SO_4$, $TNO_3$, $TCl$, $TNa$, $K^+$, $Ca^{2+}$, $Mg^{2+}$, T, and RH) was simulated to compare its effects on pH.

**a.U-ZK**

[Figure]

**b.R-PY**

[Figure]

Figure S8. Relationship between HONO and main influencing factors during (DC) the COVID-19 outbreak at U-ZK and R-PY sites. In each box, the top, middle, and bottom lines represent the 75, 50, and 25 percentiles of statistical data, respectively; the upper and lower whiskers represent the 90 and 10 percentiles of statistical data, respectively.

[Figure]

[Figure]

Figure. S9. HONO production rate through $R_1$ at U-ZK and R-PY sites before (PC) and
during (DC) the COVID-19 outbreak. In each box, the top, middle, and bottom lines
represent the 75, 50, and 25 percentiles of statistical data, respectively; the upper and
lower whiskers represent the 90 and 10 percentiles of statistical data, respectively.

[Figure]

Figure S10. Sensitivity of HONO product rate to each factor. The vertical bar represents the mean values before (PC) and during (DC) the COVID-19 outbreak. The real-time measured values of a variable and the average values of other parameters were input into the production rate of the $R_1$ reaction.

[Figure]

Figure S11. pH and $R_1$ uncertainties at the U-ZK and R-PY sites are based on two extreme scenarios of sensitivity to measurement uncertainty.

**Tables**

Table S1. Descriptions of the ten sampling sites in Henan Province, China.

| Observation sites | Classifications | Abbreviations | Coordinates | Locations | Surrounding environment |
|---|---|---|---|---|---|
| Sanmenxia | Urban site | U-SMX | 34.79 °N, 111.16 °E | Sanmenxia Environmental Protection Bureau | Roads, residential areas |
| Zhoukou | Urban site | U-ZK | 33.65° N, 114.65° E | Chuanhui District People's Government | Roads, residential areas |
| Zhumadian | Urban site | U-ZMD | 33.01° N, 114.01° E | Huanghuai College | Roads, residential areas, shopping malls |
| Xinyang | Urban site | U-XY | 32.14° N, 114.09° E | Xinyang Museum | Roads, residential areas, shopping malls |
| Anyang | Rural site | R-AY | 36.22°N, 114.39° E | Baizhuang Town Xindian North Street China Resources Gas (Andan Station) | Highways, villages, farmland |
| Xinxiang | Rural site | R-XX | 35.38° N, 114.30° E | Banzao Township Central School in Yanjin County | Villages, farmland |
| Puyang | Rural site | R-PY | 36.15° N, 115.10° E | Nanle County Longwang Temple Station | Villages, farmland |
| Jiaozuo | Rural site | R-JZ | 35.02° N, 113.35° E | The Second River Bureau of Jiefeng Village, Beiguo Township, Wuxi County | Villages, farmland |
| Shangqiu | Rural site | R-SQ | 34.56° N, 115.61° E | Liangyuan Huanghe Gudao National Forest Park | Highways, villages, farmland |
| Nanyang | Rural site | R-NY | 32.68° N, 111.70° E | Nanyang Tangshan Park | Villages, farmland |

| 273 | Table S2. The value of $\rho_s$ in other studies. | | | |
|---|---|---|---|---|
| Observation site | Period | $\rho_s$ (g/cm³) | Reference |
| Beijing | Dec 2016 | 1.4 | (Liu et al., 2017) |
| Tianjin | Dec-Jun 2015 | 1.3 | (Shi et al., 2017) |
| Xi'an | Nov-Dec 2012 | 1.4 | (Guo et al., 2017) |
| Hohhot | Winter 2015 | 1.35 | (Wang et al., 2019) |
| Northeastern USA | Feb-Mar 2015 | 1.4 | (Guo et al., 2016) |
| Crete, Greece | Aug-Nov 2012 | 1.35 | (Bougiatioti et al., 2016) |
| Alabama, USA | Jun-Jul 2013 | 1.4 | (Guo et al., 2015) |
| Georgia, USA | Aug-Oct 2016 | 1.4 | (Nah et al., 2018) |

Table S3. Summary of vehicle emission factors.

| Observation site | Period | Emission factor (%) | Reference |
|---|---|---|---|
| Beijing | 2020 | 0.79 | (Meng et al., 2020) |
| Hong Kong | 2015 | 0.4–1.8 | (Yun et al., 2017) |
| Hong Kong | 2011 | 0.5–1.6 | (Xu et al., 2015) |
| Kiesberg Tunnel | 2001 | 0.8 | (Kleffmann et al., 2003) |
| Kiesberg Tunnel | 1997 | 0.3–0.8 | (Kurtenbach and Wiesen, 2001) |
| Guangzhou | 2019 | 1.31 | (Li et al., 2021b) |

Table S4. Constants for calculating the apparent Henry's constant (H*).

[revised manuscript text omitted]

---

## Author Comment (AC2)

**No.: egusphere-2023-2913**

**Title: Measurement Report: Elevated excess-NH$_3$ can promote the redox reaction to produce HONO: Insights from the COVID-19 pandemic**

**Reviewer #1:**

**General Comments:**

In this study, the authors analyzed the chemical composition changes during the pandemic in ten urban and rural sites, and compared the HONO concentration level before and during the emission control period. The authors found that the HONO decline was relatively insignificant compared to its precursors and a detailed calculation shows that the enhanced production rate of aqueous phase reaction partially offset the effect of lower precursors. By comparing the atmospheric acids and bases concentrations, the authors suggested that the enhanced level of NH$_3$ and elevated aerosol pH due to less acidic components in the atmosphere was the reason for the higher HONO production rate. It can be one of the possible reasons, while there are several important issues that the authors did not have enough discussion or provide clear explanation. Some analysis and explanations are too simplified to give the assessment of the quality of this study.

==Thank you for your careful reading of our paper and valuable comments and==

suggestions. We believe that we have adequately addressed your comments. To facilitate your review, we used yellow highlights for your comments, green highlights for Reviewer #2, and red color indicating our own corrections in the manuscript.

**Major issues:**

1. The direct emission HONO was estimated based on the vehicle emission factors and $NO_x$ concentration level, which should reflect a general situation of normal human activities. However, during the pandemic, the emission factors could change very significantly if only necessary activities were allowed to be carried out. The authors did not mention emission profile change before and during the pandemic, which could lead to the overestimation of the effect of other pathways.

**Response:** Thanks for your comments. We determined whether it is necessary to calculate vehicle emissions, and a supplementary HONO emission factor table from vehicle emissions was added to support the selection of factors in the revised version:

"HONO can be released directly into the atmosphere through vehicle exhaust (Burling et al., 2010; Veres et al., 2010). The lifetime of HONO in the atmosphere is relatively short, so vehicle emissions significantly contribute to urban atmospheric HONO (Chen et al., 2023; Liu et al., 2021a). Considering that there has been a significant reduction in vehicle emissions in urban areas during DC. Additionally, the R-PY site is far from roads. Thus, vehicle emissions may not be the primary source of HONO for the U-ZK site during DC and R-PY sites during entire periods. To further validate the above conclusions, the conditional bivariate probability function diagrams of $NO_2$ at U-ZK and R-PY sites during PC and DC are depicted in Figure S2. $NO_2$ predominantly originated from long-distance transport at the U-ZK site during DC and the R-PY site during both PC and DC. Consequently, vehicle emissions are only calculated for the U-ZK site during the PC.

Here we use the $HONO/NO_x$ ratio to estimate HONO concentration, which is generally considered to be the vehicle emission factor (Kramer et al., 2020; Hao et al., 2020; Yu et al., 2022) for HONO. The calculation formula is as follows:

$$[HONO_{emi}] = 0.8\% \times [NO_x] \tag{1}$$

where $[HONO_{emi}]$ and $[NO_x]$ represent the HONO concentration emitted by vehicles and the observed $NO_x$ concentration, respectively. Regarding previous studies (Table S3), 0.8% was selected as the vehicle emission factor, considering differences in vehicle type, fuel composition, and other factors (Kramer et al., 2020; Hao et al., 2020; Huang et al., 2017)."

[Figure]

Figure S2. Result of conditional bivariate probability function plots: $NO_2$ at U-ZK and R-PY sites before (PC) and during (DC) the COVID-19 outbreak. The color scale bar represents $NO_2$ concentration.

Table S3. Summary of vehicle emission factors.

| Observation site | Period | Emission factor (%) | Reference |
|---|---|---|---|
| Beijing | 2020 | 0.79 | (Meng et al., 2020) |
| Hong Kong | 2015 | 0.4–1.8 | (Yun et al., 2017) |
| Hong Kong | 2011 | 0.5–1.6 | (Xu et al., 2015) |
| Kiesberg Tunnel | 2001 | 0.8 | (Kleffmann et al., 2003) |
| Kiesberg Tunnel | 1997 | 0.3–0.8 | (Kurtenbach and Wiesen, 2001) |
| Guangzhou | 2019 | 1.31 | (Li et al., 2021b) |

2. Supplement Line 107: it is very challenging to pick a representative OH concentration to represent the general situation. The authors also suggested in the introduction that OH radical concentration could change during emission control as part of atmospheric oxidizing capacity changes. While the authors did not mention such an approach in their HONO production calculation. In addition to other reaction pathways, another possibility is the change of reaction rates, like OH concentration levels and higher temperature (the authors only mentioned H and K temperature dependence but did not mention $k_1$ temperature dependence, which could be important). The authors should fully discuss the possibilities of the changes in reaction rate and possible sinks.

**Response:** Thank you for your valuable comments.

Firstly, we have modified the method for determining •OH concentration in the revised manuscript:

"•OH concentration was simulated according to the empirical model (Hu et al., 2022; Wang et al., 2025):

$$[OH] = 4.1 \times 10^9 \times \frac{J(O^1D) \times J(NO_2) \times (140 \times [NO_2] + 1) + [HONO] \times J(HONO)}{0.41 \times [NO_2]^2 + 1.7 \times [NO_2] + 1 + [NO] \times k_{NO+OH} + [HONO] \times k_{NO+OH}} \quad (12)$$

where, J ($O^1$ D), J ($NO_2$), and J (HONO) are the photolysis rates calculated using the TUV model (v5.2; available at http://cprm.acom.ucar.edu/Models/TUV/). The calculated •OH concentration varied from $0.1 \times 10^6$ to $4 \times 10^6$ molecule/cm$^3$ at U-ZK and $0.1 \times 10^6$ to $5 \times 10^6$ molecule/cm$^3$ t R-PY, which was comparable to the levels in other cities of North China (Li et al., 2018; Fuchs et al., 2017; Yang et al., 2017). Since there is no photolysis at night, the •OH concentration was assumed to be $0.8 \times 10^6$ molecule/cm$^3$ (Wang et al., 2022)."

Secondly, the sources of HONO were recalculated to better investigate the changes in HONO between PC and DC periods:

**Text S4 Sources of HONO**

4.1 Direct emission

HONO can be released directly into the atmosphere through vehicle exhaust (Burling et al., 2010; Veres et al., 2010). The lifetime of HONO in the atmosphere is relatively short, so vehicle emissions significantly contribute to urban atmospheric HONO (Chen et al., 2023; Liu et al., 2021a). Considering that there has been a significant reduction in vehicle emissions in urban areas during DC. Additionally, the R-PY site is far from roads. Thus, vehicle emissions may not be the primary source of HONO for the U-ZK site during DC and R-PY sites during entire periods. To further validate the above conclusions, the conditional bivariate probability function diagrams of $NO_2$ at U-ZK and R-PY sites during PC and DC are depicted in Figure S2. $NO_2$ predominantly originated from long-distance transport at the U-ZK site during DC and the R-PY site during both PC and DC. Consequently, vehicle emissions are only calculated for the U-ZK site during the PC.

Here we use the $HONO/NO_x$ ratio to estimate HONO concentration, which is generally considered to be the vehicle emission factor (Kramer et al., 2020; Hao et al., 2020; Yu et al., 2022) for HONO. The calculation formula is as follows:

$$[HONO_{emi}] = 0.8\% \times [NO_x] \tag{1}$$

where [$HONO_{emi}$] and [$NO_x$] represent the HONO concentration emitted by vehicles and the observed $NO_x$ concentration, respectively. Regarding previous studies (Table S3), 0.8% was selected as the vehicle emission factor, considering differences in vehicle type, fuel composition, and other factors (Kramer et al., 2020; Hao et al., 2020; Huang et al., 2017).

4.2 Homogeneous reaction of NO and •OH

The reaction between NO and •OH is the primary gas-phase reaction source of HONO at high NO concentrations, and the production rate contribution ($P_{OH+NO}$) for this reaction can be calculated as:

$$P_{OH+NO} = k_{OH+NO}[OH][NO] \tag{2}$$

where $k_{OH+NO}$ ($7.2 \times 10^{-12}$ $cm^3$ $molecule^{-1}$ $s^{-1}$) is the rate constant for the reactions at 298K (Li et al., 2012). •OH concentration was simulated according to the empirical model (Hu et al., 2022; Wang et al., 2025):

$$[OH] = 4.1 \times 10^9 \times \frac{J(O^1D) \times J(NO_2) \times (140 \times [NO_2]+1) + [HONO] \times J(HONO)}{0.41 \times [NO_2]^2 + 1.7 \times [NO_2] + 1 + [NO] \times k_{NO+OH} + [HONO] \times k_{NO+OH}} \tag{3}$$

where, $J(O^1D)$, $J(NO_2)$, and $J(HONO)$ are the photolysis rates calculated using the TUV model (v5.2; available at http://cprm.acom.ucar.edu/Models/TUV/). The cloud optical depth value for the TUV model was adjusted so that the predicted UVB radiation intensity matched the observations (Lyu et al., 2019; Wang et al., 2022b). The calculated •OH concentration varied from $0.1 \times 10^6$ to $4 \times 10^6$ molecule/$cm^3$ at U-ZK and $0.1 \times 10^6$ to $5 \times 10^6$ molecule/$cm^3$ t R-PY, which was comparable to the levels in other cities of North China (Li et al., 2018; Fuchs et al., 2017; Yang et al., 2017). Since there is no photolysis at night, the •OH concentration was assumed to be $0.8 \times 10^6$ molecule/cm$^3$ (Wang et al., 2022).

**4.3 Heterogeneous conversion of NO$_2$ to HONO**

**4.3.1 Heterogeneous dark reactions**

The heterogeneous conversion of NO$_2$ to HONO on the ground (P$_{ground}$) and on the aerosol surface (P$_{aerosol}$) was calculated based on parameters obtained from experiments or observations.

$$P_{ground} = \frac{1}{8}\gamma_1 \times [NO_2] \times C_{NO_2} \times \frac{S_g}{V} \tag{4}$$

$$P_{aerosol} = \frac{1}{4}\gamma_2 \times [NO_2] \times C_{NO_2} \times \frac{S_a}{V} \tag{5}$$

$$\frac{S_g}{V} = \frac{1}{MLH} \tag{6}$$

$$C_{NO_2} = \sqrt{\frac{8RT}{\pi M}} \tag{7}$$

where C$_{NO2}$ is the average molecular velocity of NO$_2$ molecule (m s$^{-1}$); R is the ideal gas constant; T is the temperature (K); M is the molecular weight of NO$_2$ (kg mol$^{-1}$); MLH is the height of the mixed layer, which is determined to be 50 m due to HONO formation on the ground level and its short lifetime (Liu et al., 2020b); S$_a$/V is the surface area to volume ratio of aerosol, estimated by Su et al. (Su et al., 2008).

**4.3.2 Heterogeneous photo-enhanced reactions**

The heterogeneous photo-enhanced reactions of NO$_2$ on the surface of the ground ($P_{ground + hv}$) and the surface of the aerosol ($P_{aerosol + hv}$) were calculated following (Zhang et al., 2020a):

$$P_{ground+hv} = \frac{1}{8} \times C_{NO_2} \times \frac{1}{MLH} \times \gamma_1 \times \frac{J_{NO_2}}{J_{NO_2,noon}} \times [NO_2] \qquad (8)$$

$$P_{aerosol+hv} = \frac{1}{4} \times C_{NO_2} \times \frac{S_a}{V} \times \gamma_2 \times \frac{J_{NO_2}}{J_{NO_2,noon}} \times [NO_2] \qquad (9)$$

where $J_{NO_2}$ and $J_{NO_2, noon}$ are the photolysis rate of $NO_2$ and the photolysis rate of $NO_2$ at noon during the day, respectively.

$\gamma_1$ and $\gamma_2$ are the absorption coefficient of $NO_2$ on the ground and aerosol surface, respectively, which is assumed to be $4 \times 10^{-6}$ (Yu et al., 2022; Zhang et al., 2021; Zhang et al., 2020a). Moreover, we discuss the uncertainties based on the recommended values of $2 \times 10^{-6}$–$1 \times 10^{-5}$ as upper and lower bounds(Chen et al., 2023; VandenBoer et al., 2013; Wong et al., 2011). Results show (Figure S3) that the uncertainties for $P_{ground}$, $P_{aerosol}$, $P_{groung+hv}$, and $P_{aerosol+hv}$ are −50% to 150%, −50% to 151%, −20% to 120%, and −50% to 121% at the U-ZK, respectively. At the R-PY, the uncertainties for $P_{ground}$, $P_{aerosol}$, $P_{groung+hv}$, and $P_{aerosol+hv}$ are −50% to 150%, −50% to 151%, −20% to 120%, and −50% to 121%, respectively.

**4.4 Nitrate photolysis**

The nitrate photolysis ($P_{nitrate}$) was calculated based on the measured nitrate concentration ($NO_3^-$) from $PM_{2.5}$ and nitrate photolysis rate ($J_{nitrate \rightarrow HONO}$):

$$P_{nitrate} = J_{nitrate \rightarrow HONO} \times [NO_3^-] \qquad (10)$$

where the $J_{nitrate \to HONO}$ was simulated by normalizing UV values, when the Zenit Angle is 0°, $J_{nitrate \to HONO}$ varied within the range of $1.22 \times 10^{-5}$ to $4.84 \times 10^{-4}$ s$^{-1}$, with an average value of $8.24 \times 10^{-5}$ s$^{-1}$ (Bao et al., 2018)."

Unfortunately, for MLH, $S_a/V$, and the relationship between k1 and temperature, as there were no observational data or scientifically established estimation methods, this study did not consider their variations. This omission may lead to differences in conclusions and warrants further investigation in future research.

[Figure]

Figure S2. Result of conditional bivariate probability function plots: NO$_2$ at U-ZK and

R-PY sites before (PC) and during (DC) the COVID-19 outbreak. The color scale bar represents NO$_2$ concentration.

[Figure]

Figure S3. HONO production rate using different uptake rates of $NO_2$ at the U-ZK and R-PY sites before (PC) and during (DC) the COVID-19 outbreak. (a)$P_{ground}$, (b) $P_{aerosol}$, (c) $P_{ground+hv}$, and (d) $P_{aerosol+hv}$

Table S3. Summary of vehicle emission factors.

| Observation site | Period | Emission factor (%) | Reference |
| --- | --- | --- | --- |
| Beijing | 2020 | 0.79 | (Meng et al., 2020) |
| Hong Kong | 2015 | 0.4–1.8 | (Yun et al., 2017) |
| Hong Kong | 2011 | 0.5–1.6 | (Xu et al., 2015) |
| Kiesberg Tunnel | 2001 | 0.8 | (Kleffmann et al., 2003) |
| Kiesberg Tunnel | 1997 | 0.3–0.8 | (Kurtenbach and Wiesen, 2001) |
| Guangzhou | 2019 | 1.31 | (Li et al., 2021b) |

3. It is also questionable about the contribution of $NH_3$ concentration changes to the total pH changes. Temperature, relative humidity, and other salts could also contribute to pH changes. It was not mentioned how the sensitivity tests of Line 264-275 were done and the interpretation of the results was also unclear. The authors did not give a complete pH comparison like $NH_x$ levels, only provided two sites in Figure 4. The authors mentioned the increase of pH 0.4 and 0.1 for U-ZK and R-PY sites respectively. However, based on the $NH_3$ levels shown in Table 1 and the relationship mentioned in Song et al. (2019): $\partial pHi/\partial[NH_3(g)] \approx 0.4/[NH_3(g)]$, the $NH_3$ concentration changes was only responsible for 0.13-unit pH change in U-ZK (less than half). The pH changes of most sites, if only considering $NH_3$ levels changes in Table 1, can be calculated to be around 0.1 with the exception of R-SQ where $NH_3$ concentration nearly doubled.

**Response:** Sorry for the misunderstanding. The formula in Song's study only considers the effect of $NH_3$ on the pH value of particulate matter and does not take into account other substances such as $TH_2SO_4$, $TNO_3$, T, etc., which have a greater impact on pH value. Therefore, when the $NH_3$ value in this study is brought into the formula, there is a different conclusion obtained. To explore the dominant factors that determine the high pH during the DC, we have added a detailed description of the sensitivity tests of pH to input data:

[revised manuscript text omitted]

4. Figure 4, the maximum and minimum values provided little information of the whole pH variations. A box and whisker plot are more useful to identify the general trends and variations. And there were frequent situations of maximum pH higher than 7, which could not be explained by higher $NH_3$ concentrations. Instead, it could be from the strong influence of dust components. If that situation happened frequently enough (hard to judge now based on the information given), it could be the dust components that are actually responsible for the high pH.

**Response:** Thank you for your suggestions. We redrew Figure 4 as a box diagram and replaced it in the revised version. After examining the raw data, we found that the pH data higher than 7 mainly concentrated in clean air with low pollutant concentrations. Additionally, some data had RH levels below 30%, which could lead to significant errors in the model. Thus, ISORROPIA-II was rerun only using data with RH ≥ 30% in the revised version.

[Figure]

Figure 4. Diurnal patterns of pH at ten sites before (PC) and during (DC) the COVID-19 outbreak. In each box, the top, middle, and bottom lines represent the 75, 50, and 25 percentiles of statistical data, respectively; the upper and lower whiskers represent the 90 and 10 percentiles of statistical data, respectively.

5. It should also be mentioned that the approach of the authors used to estimate AWC$_{org}$ is sensitive to the parameters chosen, such as OM/OC ratio, density, and kappa parameter. Normally, the term AWC$_{org}$ is small enough so that its influence is limited, while it is possible the uncertainty associated with the parameters chosen became big enough when inorganic salts become depleted and the relative contribution of OM got enhanced.

**Response:** Thank you for your comment. We supplemented the selection criteria for calculating parameters in the revised manuscript:

"AWC$_{org}$ is the particle water associated with the organic matters predicted using the following method:

$$\text{AWC}_{org} = \frac{m_s}{\rho_s} \frac{k_{org}}{\left(\dfrac{1}{\text{RH}} - 1\right)} \tag{2.2}$$

where $m_s$ is the mass concentration of organic matter (OM = OC × $f$). The $f$ is the conversion factor of OC, which is dependent on the extent of OM oxidation and secondary organic aerosol formation (Chow et al., 2015). Studies on the ratio of OM/OC in fourteen cities in China suggested that the mean value of $f$ was 1.59 ± 0.18 during the winter season in Northern China (Xing et al., 2013), and thus we adopted 1.6 as the $f$ in this study. $k_{org}$ is the organic hygroscopicity parameter and depends on organic functionality, water solubility, molecular weight, and oxidation level. Han et al. (2022) have reported that the $k_{org}$ generally increased with O: C ratios, with a range of 0–0.3 for 23 organics, including carboxylic acids, amino acids, sugars, and alcohols. Gunthe et al, (2011) estimated a $k_{org} = 0.06 \pm 0.01$ for the effective average hygroscopicity of the non-refractory organic particulate matter in the aerosols in Beijing. Our previous study has estimated that the uncertainties of $k_{org}$ value (0.06) for pH in the range of 0–0.3 only lead to –1–3% errors, which can be neglected (Wang et al., 2023a). Therefore, the value of 0.06 was selected in this paper. $\rho_s$ is the organic density, which was chosen to be 1.35 g/cm$^3$ following previous studies (Table S2)."

**Table S2**. The value of $\rho_s$ in other studies.

| Observation site | Period | $\rho_s$ (g/cm$^3$) | Reference |
|---|---|---|---|
| Beijing | Dec 2016 | 1.4 | (Liu et al., 2017) |
| Tianjin | Dec-Jun 2015 | 1.3 | (Shi et al., 2017) |
| Xi'an | Nov-Dec 2012 | 1.4 | (Guo et al., 2017) |
| Hohhot | Winter 2015 | 1.35 | (Wang et al., 2019) |
| Northeastern USA | Feb-Mar 2015 | 1.4 | (Guo et al., 2016) |
| Crete, Greece | Aug-Nov 2012 | 1.35 | (Bougiatioti et al., 2016) |
| Alabama, USA | Jun-Jul 2013 | 1.4 | (Guo et al., 2015) |
| Georgia, USA | Aug-Oct 2016 | 1.4 | (Nah et al., 2018) |

**Minor issues:**

6. The definition of TNHx is different in Line113 and Line 228.

**Response:** Thank you for your careful reading of our paper. The formula is used uniformly in the new version:

$$TNH_x = 17 \times \left(\frac{[NH_4^+]}{18} + \frac{[NH_3]}{17}\right)$$

7. Line 42, the study cited is the result based on a field campaign.

**Response:** Thank you for your comment. We added more references: "Nitrous acid (HONO) is a critical precursor of hydroxyl radical (OH), contributing to more than 60%

of OH production (Alicke, 2003; Platt et al., 1980; Kleffmann et al., 2005)."

8. Figure 2, the max and min as error bars provide little information about the general trends, and there are negative values.

**Response:** Thank you for your comments. We redrew Figure 2 as a box diagram and the negative values were removed after quality control.

[Figure]

Figure 2. Daily variation of $NH_3$ concentration at ten sites before (PC) and during (DC) the COVID-19 outbreak. The green dots represent the location of ten sites and their size represents the concentration of $NH_3$; In each box, the top, middle, and bottom lines represent the 75, 50, and 25 percentiles of statistical data, respectively; the upper and lower whiskers represent the 90 and 10 percentiles of statistical data, respectively.

9. Line 215, it is hard to judge if agricultural activity got weakened or not. The $NH_3$ concentration change could be due to less farm activity like less frequent animal feces cleaning, relatively higher temperature or a different regional transportation pattern.

**Response:** Thank you for your valuable suggestions. We have removed the speculation:

**Manuscript**

[revised manuscript text omitted]
 other systems (VandenBoer et al., 2014). Therefore, it is feasible to measure HONO using this instrument.

**Text S3 Detailed description of the NO₂、SO₂ and carbon analyzer.**

The $NO_2$ analyzer utilized the chemiluminescence technique to measure the concentration of $NO_2$ in the air. This involved converting $NO_2$ to NO using a molybdenum converter, and then quantifying the NO concentration. The principle behind the $SO_2$ analyzer involved measuring the amount of ultraviolet light emitted during the decay of high-energy state $SO_2$. This emitted light was used to calculate the concentration of $SO_2$.

The carbon analyzer principle is primarily based on the NIOSH-5040 method, which involves analyzing the thermal optical transmittance of quartz filter samples. It employs a calibrated non-dispersive infrared sensor to detect the evolving carbon. Under controlled conditions with inert helium gas, carbon formed during a gradually increasing temperature gradient is referred to as OC, while carbon evolved under a mixture of 90% helium.

**Text S4 Sources of HONO**

4.1 Direct emission

HONO can be released directly into the atmosphere through vehicle exhaust (Burling et al., 2010; Veres et al., 2010). The lifetime of HONO in the atmosphere is relatively short, so vehicle emissions significantly contribute to urban atmospheric HONO (Chen et al., 2023; Liu et al., 2021a). Considering that there has been a significant reduction in vehicle emissions in urban areas during DC. Additionally, the

R-PY site is far from roads. Thus, vehicle emissions may not be the primary source of

HONO for the U-ZK site during DC and R-PY sites during entire periods. To further validate the above conclusions, the conditional bivariate probability function diagrams of $NO_2$ at U-ZK and R-PY sites during PC and DC are depicted in Figure S2. $NO_2$

predominantly originated from long-distance transport at the U-ZK site during DC and the R-PY site during both PC and DC. Consequently, vehicle emissions are only calculated for the U-ZK site during the PC.

Here we use the HONO/$NO_x$ ratio to estimate HONO concentration, which is generally considered to be the vehicle emission factor (Kramer et al., 2020; Hao et al.,

2020; Yu et al., 2022) for HONO. The calculation formula is as follows:

$$[HONO_{emi}] = 0.8\% \times [NO_x] \tag{1}$$

where [$HONO_{emi}$] and [$NO_x$] represent the HONO concentration emitted by vehicles and the observed $NO_x$ concentration, respectively. Regarding previous studies (Table

S3), 0.8% was selected as the vehicle emission factor, considering differences in vehicle type, fuel composition, and other factors (Kramer et al., 2020; Hao et al., 2020; Huang et al., 2017).

4.2 Homogeneous reaction of NO and •OH

The reaction between NO and •OH is the primary gas-phase reaction source of

HONO at high NO concentrations, and the production rate contribution ($P_{OH+NO}$) for this reaction can be calculated as:

$$P_{OH+NO} = k_{OH+NO}[OH][NO] \qquad (2)$$

where $k_{OH+NO}$ ($7.2 \times 10^{-12}$ cm$^3$ molecule$^{-1}$ s$^{-1}$) is the rate constant for the reactions at

298K (Li et al., 2012). •OH concentration was simulated according to the empirical model (Hu et al., 2022; Wang et al., 2025):

$$[OH] = 4.1 \times 10^9 \times \frac{J(O^1D) \times J(NO_2) \times (140 \times [NO_2]+1) + [HONO] \times J(HONO)}{0.41 \times [NO_2]^2 + 1.7 \times [NO_2] + 1 + [NO] \times k_{NO+OH} + [HONO] \times k_{NO+OH}} \qquad (3)$$

where, J (O$^1$ D), J (NO$_2$), and J (HONO) are the photolysis rates calculated using the

TUV model (v5.2; available at http://cprm.acom.ucar.edu/Models/TUV/). The cloud optical depth value for the TUV model was adjusted so that the predicted UVB radiation intensity matched the observations (Lyu et al., 2019; Wang et al., 2022b). The calculated

•OH concentration varied from $0.1 \times 10^6$ to $4 \times 10^6$ molecule/cm$^3$ at U-ZK and 0.1 $\times$

$10^6$ to $5 \times 10^6$ molecule/cm$^3$ t R-PY, which was comparable to the levels in other cities of North China (Li et al., 2018; Fuchs et al., 2017; Yang et al., 2017). Since there is no photolysis at night, the •OH concentration was assumed to be $0.8 \times 10^6$ molecule/cm$^3$

(Wang et al., 2022).

4.3 Heterogeneous conversion of NO$_2$ to HONO

4.3.1 Heterogeneous dark reactions

The heterogeneous conversion of NO$_2$ to HONO on the ground (P$_{ground}$) and on the aerosol surface (P$_{aerosol}$) was calculated based on parameters obtained from experiments or observations.

$$P_{ground} = \frac{1}{8} \gamma_1 \times [NO_2] \times C_{NO_2} \times \frac{S_g}{V} \qquad (4)$$

$$P_{aerosol} = \frac{1}{4} \gamma_2 \times [NO_2] \times C_{NO_2} \times \frac{S_a}{V} \qquad (5)$$

$$\frac{S_g}{V} = \frac{1}{MLH} \qquad (6)$$

$$C_{NO_2} = \sqrt{\frac{8RT}{\pi M}} \qquad (7)$$

where $C_{NO2}$ is the average molecular velocity of $NO_2$ molecule (m s$^{-1}$); R is the ideal gas constant; T is the temperature (K); M is the molecular weight of $NO_2$ (kg mol$^{-1}$); MLH is the height of the mixed layer, which is determined to be 50 m due to HONO formation on the ground level and its short lifetime (Liu et al., 2020b); $S_a/V$ is the surface area to volume ratio of aerosol, estimated by Su et al. (Su et al., 2008).

4.3.2 Heterogeneous photo-enhanced reactions

The heterogeneous photo-enhanced reactions of $NO_2$ on the surface of the ground ($P_{ground + h\nu}$) and the surface of the aerosol ($P_{aerosol + h\nu}$) were calculated following (Zhang et al., 2020a):

$$P_{ground+h\nu} = \frac{1}{8} \times C_{NO_2} \times \frac{1}{MLH} \times \gamma_1 \times \frac{J_{NO_2}}{J_{NO_2,noon}} \times [NO_2] \qquad (8)$$

$$P_{aerosol+h\nu} = \frac{1}{4} \times C_{NO_2} \times \frac{S_a}{V} \times \gamma_2 \times \frac{J_{NO_2}}{J_{NO_2,noon}} \times [NO_2] \qquad (9)$$

where $JNO_2$ and $JNO_{2, noon}$ are the photolysis rate of $NO_2$ and the photolysis rate of $NO_2$ at noon during the day, respectively.

$\gamma_1$ and $\gamma_2$ are the absorption coefficient of $NO_2$ on the ground and aerosol surface, respectively, which is assumed to be $4 \times 10^{-6}$ (Yu et al., 2022; Zhang et al., 2021; Zhang et al., 2020a). Moreover, we discuss the uncertainties based on the recommended values of $2 \times 10^{-6}$–$1 \times 10^{-5}$ as upper and lower bounds(Chen et al., 2023; VandenBoer et al.,

2013; Wong et al., 2011). Results show (Figure S3) that the uncertainties for $P_{ground}$,

$P_{aerosol}$, $P_{groung+hv}$, and $P_{aerosol+hv}$ are −50% to 150%, −50% to 151%, −20% to 120%, and

−50% to 121% at the U-ZK, respectively. At the R-PY, the uncertainties for $P_{ground}$,

$P_{aerosol}$, $P_{groung+hv}$, and $P_{aerosol+hv}$ are −50% to 150%, −50% to 151%, −20% to 120%, and

−50% to 121%, respectively.

4.4 Nitrate photolysis

The nitrate photolysis ($P_{nitrate}$) was calculated based on the measured nitrate concentration ($NO_3^-$) from $PM_{2.5}$ and nitrate photolysis rate ($J_{nitrate \rightarrow HONO}$):

$$P_{nitrate} = J_{nitrate \rightarrow HONO} \times [NO_3^-] \tag{10}$$

where the $J_{nitrate \rightarrow HONO}$ was simulated by normalizing UV values, when the Zenit Angle is 0°, $J_{nitrate \rightarrow HONO}$ varied within 
[revised manuscript text omitted]

| Observation site | Period | $\rho_s$ (g/cm$^3$) | Reference |
|---|---|---|---|
| Beijing | Dec 2016 | 1.4 | (Liu et al., 2017) |
| Tianjin | Dec-Jun 2015 | 1.3 | (Shi et al., 2017) |
| Xi'an | Nov-Dec 2012 | 1.4 | (Guo et al., 2017) |
| Hohhot | Winter 2015 | 1.35 | (Wang et al., 2019) |
| Northeastern USA | Feb-Mar 2015 | 1.4 | (Guo et al., 2016) |
| Crete, Greece | Aug-Nov 2012 | 1.35 | (Bougiatioti et al., 2016) |
| Alabama, USA | Jun-Jul 2013 | 1.4 | (Guo et al., 2015) |
| Georgia, USA | Aug-Oct 2016 | 1.4 | (Nah et al., 2018) |

**Table S3. Summary of vehicle emission factors.**

| Observation site | Period | Emission factor (%) | Reference |
|---|---|---|---|
| Beijing | 2020 | 0.79 | (Meng et al., 2020) |
| Hong Kong | 2015 | 0.4–1.8 | (Yun et al., 2017) |
| Hong Kong | 2011 | 0.5–1.6 | (Xu et al., 2015) |
| Kiesberg Tunnel | 2001 | 0.8 | (Kleffmann et al., 2003) |
| Kiesberg Tunnel | 1997 | 0.3–0.8 | (Kurtenbach and Wiesen, 2001) |
| Guangzhou | 2019 | 1.31 | (Li et al., 2021b) |

**Table S4. Constants for calculating the apparent Henry's constant (H*).**

[revised manuscript text omitted]

---

## Author Response (AR2)

No.: egusphere-2023-2913

**Title: Measurement Report: Elevated excess-NH$_3$ can promote the redox reaction to produce HONO: Insights from the COVID-19 pandemic**

**Comments:**

First of all, I agree that the authors have made a great improvement in the modelling approach and the results interpretation. My concerns for the uncertainties of HONO production and NH$_3$ impact have been largely addressed. However, I still need to point out that the focus of this study mentioned in abstract, conclusion and even title is a very specific mechanism that, based on the output, cannot be fully confirmed. For example, based on Table S9 and Figure S7, the pH difference of PC and DC was less than half unit for most of the sites, some are very close; the effect of temperature seems to have a more significant impact on pH value compared to other factors. As the effect of temperature, the authors stated in Line 282-289 that temperature had no obvious impact on HONO soil emission. However, it is hard to judge based on the temperature – HONO relationship if soil emission was only part of the total budget. The temperature difference between PC and DC (Table S7) can be regarded as around and way above freezing point, respectively, that could lead to a certain level of difference for soil emission potential, soil water content, soil pH, etc [ref1], which could be an important factor for HONO budget difference. It will be hard to judge the contributions of different mechanisms when there are other unknown sources existing simultaneously. Therefore, I would encourage the authors to re-assess the focus of this study, and avoid overstating the importance of one specific mechanism when there are too many other uncertainties.

Thank you for your careful reading of our paper and valuable comments and suggestions. We believe that we have adequately addressed your comments. To facilitate your review, we used green highlights for your comment and red color indicating our own corrections in the manuscript.

Firstly, sorry for the misunderstanding regarding the dominant factor influencing the increase in particle pH during the DC period. We have supplemented and refined the discussion in the Discussion, Abstract, and Conclusion sections.

**Results and discussion**

According to the average values of input data during PC (Blue line in Fig. S8) and DC (Red line in Fig. S8) at U-ZK and R-PY sites respectively, the changes in pH ($\Delta$pH in Fig. 5) indicate that the decrease in $TNH_x$ concentration and the increase in T in DC led to a decrease in pH values ($\Delta$pH: 0.09 at U-ZK and 0.08 at R-PY sites) compared to PC. However, this effect was outweighed by the decrease in $TH_2SO_4$ ($\Delta$pH: 0.07 and 0.8 at U-ZK and R-PY sites, respectively) and $TNO_3$ ($\Delta$pH: 0.05 and 0.4 at U-ZK and R-PY sites, respectively) concentrations as well as the increase in $K^+$ ($\Delta$pH: 0.03 at U-ZK and 0.2 at R-PY site) and $Mg^{2+}$ ($\Delta$pH: 0.01 at U-ZK and 0.04 at R-PY site) concentrations in the DC, and resulting in an overall increase in pH values in the DC. Furthermore, the relationship between particle pH with the concentrations of Required-$NH_x$, and Excess-$NH_x$, which considers all chemical components, is investigated to examine the dominant factor on the increasing pH in DC. As shown in Fig. 6, the higher Excess-$NH_x$ concentrations in the DC led to higher increases in pH values ($\Delta$pH: 1 at U-ZK and 0.5 at R-PY site) than those in PC ($\Delta$pH: 0.3 at U-ZK and 0.2 at R-PY site), thus Excess-$NH_x$ concentrations may be the key factor in promoting the pH values.

**Abstract**

"Sensitivity analysis indicated that the decrease in anion concentrations (especially sulfate and nitrate) and the increase in cation concentrations during the COVID-19 pandemic led to an increase in particle pH. In other words, the excess ammonia determined the promoting pH."

**Conclusions**

"Furthermore, under the environmental conditions of increased anion concentrations (especially sulfate and nitrate) and increased cation concentrations, the pH values increased by 0.5 and 0.3 at U-ZK and R-PY increased during the pandemic, respectively."

Secondly, through literature review and calculations, we found that the contribution of soil emissions to HONO under low temperature conditions (below 10°C) during the observation period can be neglected. We have added this information in Section 2.3.2 on the sources of HONO.

"Soil emission has been demonstrated to be a major source of HONO, which is affected by temperature to some extent (Liu et al., 2020b;Liu et al., 2020a). However, during the sampling periods, there was no significant positive correlation between HONO concentration and temperature (Fig. S4). In addition, temperatures did not exceed 10°C, under which the soil HONO emission rate is generally considered to be zero (Zhang et al., 2023). Furthermore, the equilibrium gas-phase concentration over an aqueous solution of nitrous acid, [HONO]*, a key parameter controlling the exchange of HONO between the gas and aqueous phase in soil, is calculated according to Su et al. (2011)The results indicate that the temperature difference between PC and DC periods only led to approximately a 0.01% concentration change. On the other hand, studies on the sources of HONO in the North China Plain of China during winter consistently showed that soil HONO emissions contribute around 1%(Zhang et al., 2023;Liu et al., 2020a;Liu et al., 2020b). Therefore, this study does not consider soil HONO emissions."

Lastly, we also recognize that further research is needed to support the conclusions regarding the generation of HONO from redox reactions. Therefore, we have added a discussion on the limitations of the calculation methods and conclusions in the revised manuscript, and we have modified the Abstract and Title accordingly.

Considering the conclusions of this study are based solely on observational data, there are certain limitations. For example, only the changes in the $R_1$ reaction of $PM_{2.5}$ were calculated, without considering variations in components, pH values, and $R_1$ reaction rates of coarse particles. Additionally, although this study selected scenarios with RH > 60% to calculate the $R_1$ reaction to ensure the presence of a liquid phase, it is evident that this approach overlooks some $R_1$ reactions. Furthermore, due to thermodynamic model calculations of pH values, changes in the mixed state of particle components,

and the omission of organic acids, alongside the absence of gaseous $HNO_3$ and $HCl$ in this study, these factors may lead to inaccuracies in pH value simulations and uncertainty in $R_1$ calculations(Pye et al., 2020;Haskins et al., 2018;Nah et al., 2018). Therefore, there is a certain degree of uncertainty in the conclusions regarding the growth of $R_1$ reactions in this paper. Nevertheless, by calculating the changes in $R_1$ reactions, this study provides a possible explanation for the relatively small decrease in HONO during the epidemic period.

**Title**

"Measurement Report: Elevated atmospheric ammonia may promote the particle pH and HONO formation: Insights from the COVID-19 pandemic"

**Abstract**

"The calculation of reaction rates indicates that during the epidemic, the increase in pH may promote the generation of HONO by facilitating redox reactions, which highlights the importance of coordinating the control of $SO_2$, $NO_x$, and $NH_3$ emissions."

Line 295-297: This is a very ideal situation that is based on the assumption of fully internal mixing of PM components. For example, with the reduced level of SNA, there could be more externally mixed dust particles that have higher pH values and also behave as better temporal sink for HONO [ref2].

**Response:** Thanks for your suggestion. We have added a discussion on the limitations of the calculation methods and conclusions in the revised manuscript.

Considering the conclusions of this study are based solely on observational data, there are certain limitations. For example, only the changes in the $R_1$ reaction of $PM_{2.5}$ were calculated, without considering variations in components, pH values, and $R_1$ reaction rates of coarse particles. Additionally, although this study selected scenarios with RH > 60% to calculate the $R_1$ reaction to ensure the presence of a liquid phase, it is evident that this approach overlooks some $R_1$ reactions. Furthermore, due to thermodynamic model calculations of pH values, changes in the mixed state of particle

components, and the omission of organic acids, alongside the absence of gaseous $HNO_3$ and HCl in this study, these factors may lead to inaccuracies in pH value simulations and uncertainty in $R_1$ calculations(Pye et al., 2020;Haskins et al., 2018;Nah et al., 2018). Therefore, there is a certain degree of uncertainty in the conclusions regarding the growth of $R_1$ reactions in this paper. Nevertheless, by calculating the changes in $R_1$ reactions, this study provides a possible explanation for the relatively small decrease in HONO during the epidemic period.

Line 55-56, the reduction of HONO seems not significantly different than that of NO2 value. Considering that there is already wet surface production mechanism of HONO [ref3], is there any potential artifact for the surface production of HONO on wet liquid surface of MARGA sampling inlet? Has the relevant quality control been performed to verify the influence?

**Response:** Thanks for your comment.

To highlight the difference in HONO and NOx concentration reductions, we further supplemented the description: "Liu et al. (2020a) observed that the decrease in HONO concentration during the pandemic period was only 31% (from 1.5 ppb to 0.9 ppb), which was significantly lower than the reductions in NO (62%, from 26.3 to 4.2 ppb) and $NO_2$ (36%, from 15.5 to 6.2 ppb)."

Yes, I agree with your opinion, the use of the wet-flow diffusion tube method by MARGA can result in the generation of HONO from $NO_2$, which is a limitation of the instrument. However, a significant decrease in $NO_2$ during DC should also lead to a corresponding decrease in surface production of HONO, at least it will not promote HONO generation during the DC period.

Figure 7&8, the results of the uncertainties analysis done in Text S4 have not been incorporated into these two figures and the relevant discussions.

**Response:** Thanks for your suggestion. We added two figures (Figures S9 and S10, two extreme scenarios) to illustrate the impact of the uncertainty in HONO calculations on HONO sources.

"Moreover, all the known HONO production sources rates including $P_{emi}$, $P_{OH + NO}$, $P_{ground}$, $P_{ground+hv}$, $P_{aerosol}$, $P_{aerosol+hv}$, and $P_{nitrate}$ (Fig. 7, Fig S9 and S10) show a decreasing trend from PC to DC, with the total reductions of 38% (from 30% to 45% in the scenario with the minimum and maximum uncertainty, respectively) and 79% (from 77% to 82% in the scenario with the minimum and maximum uncertainty, respectively) for U-ZK and R-PY sites, respectively."

[Figure]

Figure S9. Maximum uncertainty values for HONO sources at U-ZK and R-PY sites were compared between the pre-COVID-19 outbreak (PC) and during the COVID-19 (DC). Refer to Text S4 for details on the calculation methods.

[Figure]

Figure S10. Minimum uncertainty values for HONO sources at U-ZK and R-PY sites were compared between the pre-COVID-19 outbreak (PC) and during the COVID-19 (DC). Refer to Text S4 for details on the calculation methods.

We incorporated the instrument's measurement uncertainties for $NO_2$ and HONO as well as calculation uncertainties for $R_1$ into Fig.8. The shadows in the figure represent the uncertainties of $NO_2$ measurement (±10%), HONO measurement (±20%), and the HONO formation rate of $R_1$ reaction (–78–123%), respectively.

"Even considering the above uncertainty in Fig. 8, it can still be observed that during the DC period, the decrease in HONO was less than that of $NO_2$, and the rate of the $R_1$ reaction increased."

[Figure]

Figure 8. Decline ratios of a. $NO_2$, b. HONO concentration, and c. HONO production rate at U-ZK and R-PY sites before (PC) and during (DC) the COVID-19 outbreak. The center point represents the mean value, and the upper and lower whiskers represent the 95% confidence interval of the mean. The shadows in the figure represent the uncertainties of $NO_2$ measurement (±10%), HONO measurement (±20%), and the HONO formation rate of $R_1$ reaction (−78–123%), respectively.